# Computational estimates of mechanical constraints on cell migration through the extracellular matrix

**Ondrej Maxian**[1], **Alex Mogilner**[1,2], **Wanda Strychalski**[3]*

**1** Courant Institute of Mathematical Sciences, New York University, New York, New York, United States of America, **2** Department of Biology, New York University, New York, New York, United States of America, **3** Department of Mathematics, Applied Mathematics and Statistics, Case Western Reserve University, Cleveland, Ohio, United States of America

* wis6@case.edu

**Data Availability Statement:** The data underlying the results presented in the study are available from Github (https://github.com/omaxian/CellMotility).

## Abstract

Cell migration through a three-dimensional (3D) extracellular matrix (ECM) underlies important physiological phenomena and is based on a variety of mechanical strategies depending on the cell type and the properties of the ECM. By using computer simulations of the cell's mid-plane, we investigate two such migration mechanisms—'push-pull' (forming a finger-like protrusion, adhering to an ECM node, and pulling the cell body forward) and 'rear-squeezing' (pushing the cell body through the ECM by contracting the cell cortex and ECM at the cell rear). We present a computational model that accounts for both elastic deformation and forces of the ECM, an active cell cortex and nucleus, and for hydrodynamic forces and flow of the extracellular fluid, cytoplasm, and nucleoplasm. We find that relations between three mechanical parameters—the cortex's contractile force, nuclear elasticity, and ECM rigidity—determine the effectiveness of cell migration through the dense ECM. The cell can migrate persistently even if its cortical contraction cannot deform a near-rigid ECM, but then the contraction of the cortex has to be able to sufficiently deform the nucleus. The cell can also migrate even if it fails to deform a stiff nucleus, but then it has to be able to sufficiently deform the ECM. Simulation results show that nuclear stiffness limits the cell migration more than the ECM rigidity. Simulations show the rear-squeezing mechanism of motility results in more robust migration with larger cell displacements than those with the push-pull mechanism over a range of parameter values. Additionally, results show that the rear-squeezing mechanism is aided by hydrodynamics through a pressure gradient.

## Author summary

Computational simulations of two different mechanisms of 3D cell migration in an extracellular matrix are presented. One mechanism represents a mesenchymal mode, characterized by finger-like actin protrusions, while the second mode is more amoeboid in that rear contraction of the cortex propels the cell forward. In both mechanisms, the cell generates a thin actin protrusion on the cortex that attaches to an ECM node. The cell is then

**Funding:** This work was supported by the a grant from the Simons Foundation (https://www.simonsfoundation.org) [429808] to WS, the US Army Research Office grant (https://www.arl.army.mil) [W911NF-17-1-0417] to AM, the National Science Foundation (US) Graduate Research Fellowship (http://www.nsf.gov) [DGE-1342536] to OM, and by the Henry MacCracken fellowship (https://gsas.nyu.edu/content/nyu-as/gsas/admissions/financial-aid/graduate-school-fellowships-and-assistantships.html) to OM. The funders had no role in study design, data collection and analysis, decision to publish, or preparation of the manuscript.

**Competing interests:** The authors have declared that no competing interests exist.

either pulled (mesenchymal) or pushed (amoeboid) forward. Results show both mechanisms result in successful migration over a range of simulated parameter values as long as the contractile tension of the cortex exceeds either the nuclear stiffness or ECM stiffness, but not necessarily both. However, the distance traveled by the amoeboid migration mode is more robust to changes in parameter values, and is larger than in simulations of the mesenchymal mode. Additionally, cells experience a favorable fluid pressure gradient when migrating in the amoeboid mode, and an adverse fluid pressure gradient in the mesenchymal mode.

## Introduction

The ability of cells to navigate a complex three-dimensional (3D) extracellular matrix (ECM) is essential in the physiology of health and disease. One example of a process important for health is fibroblasts moving through the ECM to heal wounds [1]. On the other hand, one of the hallmarks of cancer is the migration of metastatic cancer cells across the ECM [2]. More often than not, cells move in cohesive groups, but many physiological phenomena involve single cell migration [3], which is our focus here. Understanding the mechanics of this migration is a great present challenge which can be met by combining cell biological, biophysical, and computational approaches [4].

There are a bewildering number of observed mechanical strategies of cell locomotion in 3D, reflecting the complexity and adaptability of the cell's mechanical modules. The most frequently described migration modes are mesenchymal and amoeboid [5], but distinctions between these two modes are not clear-cut. In the mesenchymal mode, cells are elongated and polarized, with protrusion activity at the short front, retraction activity at the short rear and opposite end, and tensed long sides. In this mode, integrin-dependent adhesions are distributed all over the cell surface and are crucial for migration, as inhibition of integrin stops the motion of the cells [6]. Cells protrude at the front, form a pseudopod, attach it firmly to ECM fibers, and generate contraction within the pseudopod [7] and several microns behind its tip [8]. The pseudopod can be a cylindrical lobopod in a stiff ECM or a branched, finger-like lamellipodium in a soft matrix [6]. After the protrusion and contraction are generated, a continuous release of adhesions at the rear results in translocation of the cell body forward.

In the amoeboid mode, cells are less polarized and have a more rounded shape. They have a more uniform distribution of cytoskeletal structures and/or membrane blebs around the periphery [5, 6] and migrate by squeezing through the pores of the ECM. One example of amoeboid migration is an epithelial cell moving through a 3D collagen matrix [9], where the nucleus is observed leading the cell front. The contractile cell body trails behind, and actomyosin contraction propels the nucleus forward, driving the migration of the cell. It appears from the images in [9] that the cells generate ECM deformations which store elastic energy. The cell propels forward when the elastic ECM deformations are released [10]. Another example of cells harnessing ECM deformations to create locomotion was recently reported in [11].

The modes of migration are malleable: cells are able to switch from one mode to another depending on the physical and geometric properties of the cell and ECM [5, 6, 12]. What unites these modes is that the cell's migration in the ECM depends crucially on myosin-powered contraction [4], which is not the case for cell migration on flat 2D surfaces [13]. In 2D, the cell's largest organelle, the nucleus, rides effortlessly atop the actomyosin locomotory network [14], while inside the 3D ECM the problem of moving the bulky nucleus through the matrix becomes the center of the cell's mechanical effort [4, 15]. Recent work indicates that the steric

resistance the 3D matrix presents against the forward propulsion of the nucleus is a universal constraint for 3D cell migration [16–18]. Myosin-generated forces are critical for overcoming this resistance.

Computational modelling is a valuable complement to experiments in the understanding of complex cell migration mechanics. Modeling of cell motility on flat 2D surfaces is very advanced due to a relative simplicity of 2D cell motile appendages that are amenable to description in terms of partial differential equations [19]. 3D cell migration and respective modeling are much more complex. In 2005, Zaman et al. proposed one of the first (highly simplified) force balance models for 3D cell migration, which showed that migration progress depends on adhesion strength and the ECM's mechanical and geometric properties [20]. One of the subsequent models included a continuum approach, in which each modelled cell was simplified as a self-protrusive 3D elastic unit interacting with an elastic ECM through detachable bonds [21]. Other continuous mechanical models focused on the cell shape, rather than its interactions with its deformable environment [22–24]. A few detailed, agent-based models of migrating cells immersed in a deformable ECM included models of the ECM, cell cortex, and membrane using networks of viscoelastic links [25, 26]. Another very recent effort investigated the influence of the flow of interstitial fluid on the cell's migration through the ECM [10].

A few recent models started to investigate specifically the influence of the nucleus on 3D cell migration. Sakamoto et al. [27] proposed a computational model that took into account the viscoelastic properties of the cell body. By using a finite-element method and prescribing cyclic protrusion of the leading edge of the cell, the authors predicted that the mesenchymal-to-amoeboid transition is caused by a reduced adhesion and an increased switching frequency between protrusion and contraction. A 2D mechanical model to simulate the migration of a HeLa cell with a large deformable cell body through a micro-channel was proposed in [23]. The cellular Potts modeling framework is especially well suited for modeling mechanical aspects of cell migration in a 3D environment, and this approach has led to visually stunning simulations of the deforming nucleus in cells crawling through the ECM [28]. A very detailed model of a glioma cell represented by two elastic circular curves, an inner curve corresponding to the nucleus of the cell and an outer curve corresponding to the cell basal membrane, was proposed in [29]. In this model, migration of the glioma cell through a tissue made of normal cells (also represented by elastic curves) was simulated with the immersed boundary method, with viscous fluid mediating interactions of the membranes and cortices. Last, but not least, a comprehensive mathematical model based on an energy minimization approach was used to investigate cell movements inside a channel composed of ECM [30]. Simulations of this model reproduced deformations of the elastic, initially spherical nucleus into an elongated shape able to squeeze through the channel.

These models made inroads into the general question of what the mechanical constraints imposed by squeezing the nucleus through the ECM are on cell migration, yet the following specific questions remain open: Given nuclear and ECM mechanical properties, how strong should the myosin-powered contraction be to generate persistent cell locomotion? Could one or the other locomotory mode have a mechanical advantage? What are the migration mechanics if the nucleus is much stiffer than the ECM, or vice versa? What are the mechanical roles of the nucleoplasm, cytoplasm and interstitial fluid?

In this study, we model a two-dimensional cross section of the cell as two elastic contours. One represents the nucleus, the diameter of which is sometimes on the order of 10 $\mu$m [17, 31]. The other contour, which envelops the nucleus, represents the cell cortex. The cortex has contractile and protrusive activities, while the nucleus does not. In some cases, the nucleus is almost as great in size as the whole cell [31]. In this case, which we focus on, the challenge of

propelling the nucleus through the ECM is central for cell migration. We also consider the case when the nucleus is significantly smaller than the cell. We model the ECM as an elastic network. Viscous fluids permeate the ECM (interstitial fluid) and fill the spaces inside the nucleus (nucleoplasm) and between the cortex and nucleus (cytoplasm).

We model two mechanical strategies of cell migration: in one, resembling a mesenchymal mode, the cortex makes a protrusion, the tip of which adheres to an ECM node, and then global cortex contraction pulls the cell forward. In another, resembling one of the amoeboid modes, the cortex makes two attachments to ECM nodes and contracts only at one side, pushing the nucleus at that side through the ECM gaps at the opposite side. In order to simulate the coupled fluid-structure interaction problem, the model is formulated using the method of regularized Stokeslets [32]. In this method, the main force balance includes the elastic and contractile forces from the cell and ECM along with viscous fluid stresses. The fluid velocity computed by the method determines the dynamics of cell migration. We find that the migration is successful if the contractile tension of the cell cortex is greater than the characteristic force needed to deform either the nucleus or ECM, but not necessarily both. Simulations reproduce the observed amoeboid and mesenchymal morphodynamics and predict that the amoeboid mode overall performs better mechanically than the mesenchymal mode.

## Materials and methods

### Qualitative model description

To avoid the computational expense of 3D simulations, we consider a planar cross section of the cell and a cross section of the ECM in the same plane around the cell. Essentially, we approximate the dynamics as occuring in the mid-plane of the cell. This approximation is not rigorous, so the model is, strictly speaking, 2D, but it captures the essential 3D effect of squeezing the deformable cell through the deformable ECM.

In short (mathematical and computational details are described in the following subsections and in S1 Text), the simulated cell consists of an elastic and protrusive-contractile contour, representing the cell's actomyosin cortex and membrane. Inside the cell is a deformable nucleus represented by another elastic contour. The nucleus and cortex have an elastic stretching energy, which is a conventional modeling choice [29, 33]; how well this approximation represents the cell mechanics is an open question. For example, other models of the cell have included a bulk elasticity throughout the cytoplasm [26, 34], which would be an extension of the model presented here. We also include a contour bending energy in some simulations (see S1 Text), although the latter has little effect on the dynamics in most parameter regimes. The entire cell, both nucleus and cortex, is embedded into an ECM represented by a 2D elastic node-spring network (shown in Fig 1).

The cell and ECM network are immersed in an incompressible, viscous fluid. This implies that the cell and nucleus are filled with fluid, with a constant volume (area) of the nucleoplasm inside the nucleus and a constant volume of the cytosol between the nucleus and cortex. For simplicity, the viscosities of the nucleoplasm, cytoplasm and interstitial fluid are the same, and both the cortex and nuclear membranes are assumed impermeable to fluids, i.e., each satisfies a no-slip boundary condition at the fluid/contour interface. The fluid incompressibility combined with the assumption of impermeability means the 2D cell area is approximately conserved over time, and there is no flow across the cortex/nucleus boundary.

The fundamental force balance in the model is implemented as follows: a net elastic/protrusive/contractile force at each node of the nucleus, cortex and ECM is applied to the fluid. The fluid's velocity and pressure are then calculated by solving the Stokes equations using the

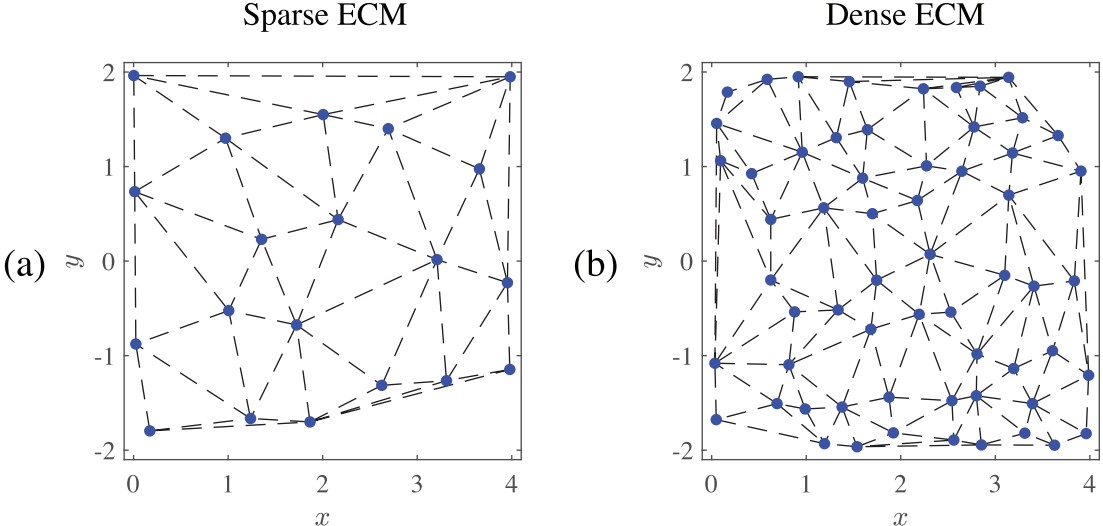

**Fig 1. Sample ECMs used for cell motility simulations.** The ECMs are generated on the box $(x, y) \in [0, 4] \times [-2, 2]$ (lengthscale units are in cell diameter). Blue points show the network, and black dotted lines show the spring connections The sparse network in (a) has 20 nodes, and the dense network in (b) has 60 nodes.

method of regularized Stokeslets [32]. This method relies on the linearity of the Stokes equations and involves using the free space Green's function for Stokes flow to compute the velocity and pressure distribution generated by a collection of regularized point forces. The regularization parameter, $\epsilon$, controls the width of the force regularization. Explicit formulas for the velocity and pressure are given in Section S1 of S1 Text. Because all of the nodes move with the computed fluid velocity, the ECM nodes do not penetrate the cortex, and the nodes of the cortex do not penetrate the nucleus up to discretization errors.

We model two motile strategies. One of them loosely resembles mesenchymal locomotion, during which cells migrate by forming a long, finger-like protrusion from the cell body into the ECM and generate traction forces behind the tip of the protrusion [8]. In the model, the cell first generates a finger-like protrusion that adheres to an ECM node. Then, global contraction of the cortex pulls the nucleus and cortex forward, ending the motile cycle. Fig 2I shows a schematic of this mode.

Another motile strategy loosely resembles one of the amoeboid modes. In this mode, the cortex first makes two protrusions that attach to two ECM nodes, and the half of the cortex behind the nucleus contracts. This cortex contraction squeezes the nucleus forward through the ECM network [9, 16, 35]. Fig 2II shows a schematic of this mode.

We now describe the model equations governing mechanics of the ECM, cortex and nucleus, as well as the dynamic processes of the two motility mechanisms. The fluid mechanics that couple the solid deformations of the ECM, cortex, and nucleus to the fluid are described in Section S1 of S1 Text.

## Extracellular matrix

Our model ECM is composed of elastic fibers that are immersed in fluid and linked by virtual springs. We use the term "virtual" to indicate that the springs connecting the nodes do not sterically inhibit cell migration. Rather, we assume that cells interact only with the ECM nodes, and that the springs between the ECM nodes only provide restoring forces to the nodes. Thus, we treat the ECM simply as a collection of point-nodes and form a lattice by triangulating

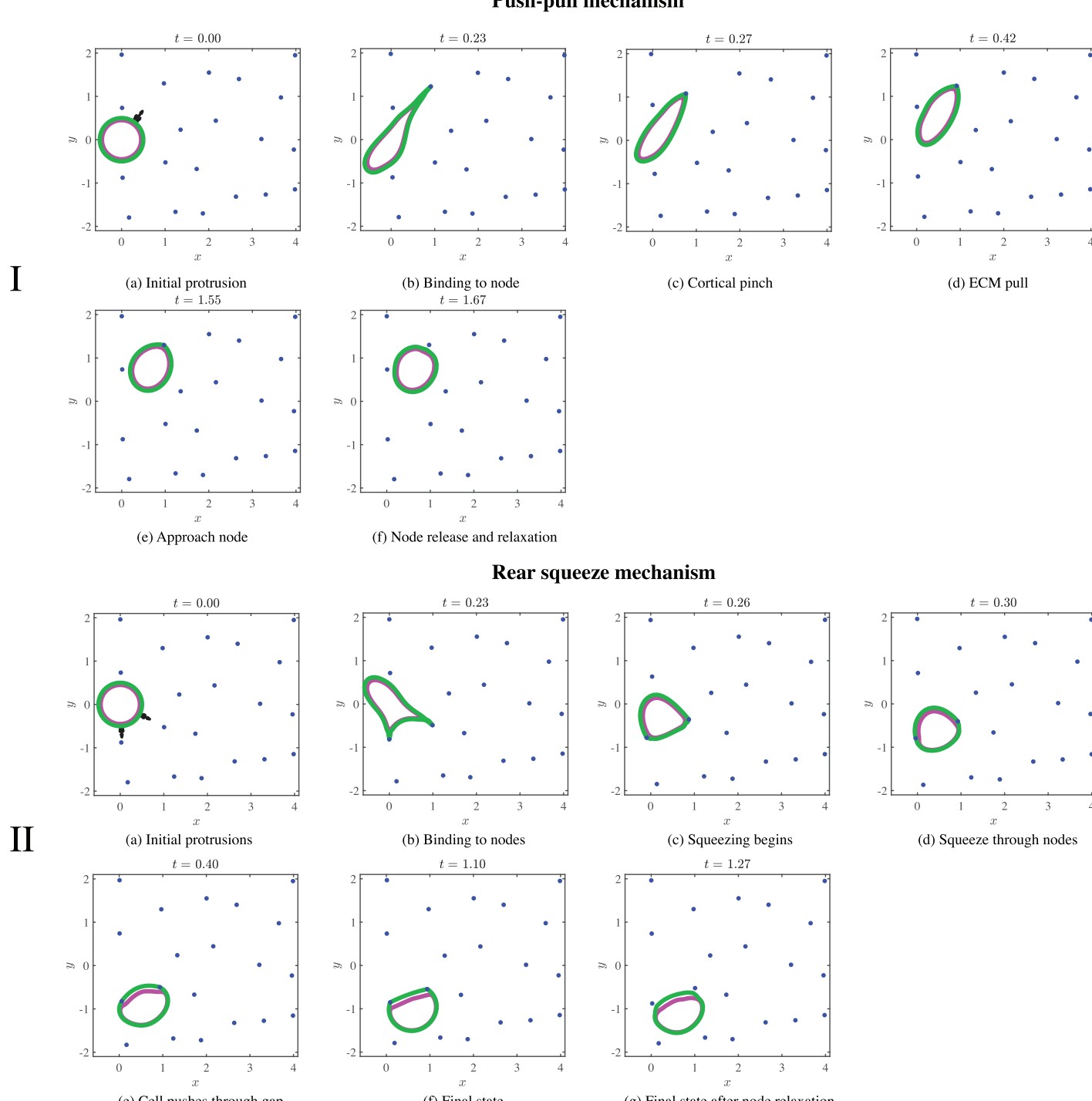

**Fig 2. Stages of motility for the push-pull and rear-squeezing mechanisms of motility.** The position of the cortex (green), nucleus (magenta), ECM nodes (blue), and applied forces (black arrows) are shown at several times values increasing from left to right and up to down. For panels in I, a random protrusion appears on the cortex (a) and extends until it comes into contact with an ECM node in (b). The cortex then becomes extremely stiff, which initially pulls the ECM node towards the cell in (c) and (d). Forces due to ECM elasticity eventually balance cellular forces to result in translocation of the entire cell (e). After coming to rest close to the node, the cell releases the node and the node moves away (f). The system is then allowed to re-equilibrate. Panels in II show the stages of motility when the cell uses rear contraction. Two protrusions form on the front (right) edge of the cell cortex in (a), then extend until both come into contact with ECM nodes in (b). The cortex is then separated into two regions. The leading edge region between the attachments remains loose, while the longer rear region behind the attachments contracts. Squeezing of the cell through the ECM gap occurs as the rear cortex contracts (c–e). The cell passes through the gap between the two nodes in (e), then comes to rest (f) when the region of the cortex between the two nodes is short and stiff. Next, the cell releases the ECM nodes, which move back toward their original positions in the final resting state (g), and the cortex relaxes. This completes one motility cycle.

these points. Any two nodes that share an edge (in the triangulation) are linked together by a virtual spring. Suppose that $N(i)$ is the set of neighbors for the $i$th fiber. Then the force at ECM node $i$ can be computed as

$$\hat{\boldsymbol{F}}_i^{\mathrm{ECM}} = \hat{\boldsymbol{F}}_i^{\mathrm{ECM},0}(\boldsymbol{X}_i) - k^{\mathrm{ECM}} \sum_{j \in N(i)} (\boldsymbol{X}_i - \boldsymbol{X}_j). \tag{1}$$

In Eq (1), $\hat{\boldsymbol{F}}_i^{\mathrm{ECM},0}$ is the unique "pinning-down" force that ensures the fibers are motionless at the beginning of the simulation. Since we construct random lattices, there will be a net force initially on each fiber in the absence of $\hat{\boldsymbol{F}}^{\mathrm{ECM},0}$ simply because the points are not located on a regularly spaced mesh (see Fig 1). $\hat{\boldsymbol{F}}^{\mathrm{ECM},0}$ penalizes translations of the lattice while ensuring that the fibers do not collapse onto each other during a dynamic simulation. This force can also be thought of as providing a spring rest length, and is calculated in practice by precomputing a reference location to which we tether the point $\boldsymbol{X}_i$. The second term in Eq (1) describes the elastic contributions of the springs with stiffness $k^{\mathrm{ECM}}$ that connect fiber $i$ to its neighbors. The ECM effectively behaves like a viscoelastic material with a spring and dashpoint in parallel (Kelvin-Voigt material). Viscosity arises from the fluid, and the elasticity comes from the forces in Eq (1).

We normalize lengths and distances so that the cell has diameter 1. The ECM nodes are therefore generated on the two dimensional box $(x, y) \in [0, 4] \times [-2, 2]$ (excluding a region near the origin where the cell is initially placed). Fig 1 shows the ECMs used in our numerical experiments. We use 20 nodes for a sparse ECM and 60 nodes for a dense ECM. The black dotted lines show the elastic lattice that connects the nodes. Fig 1 shows that the 20 and 60 node ECMs have average fiber spacings of about 1.5 and 0.5 cell diameters, respectively. This means that the ECM node spacing is either larger (for a sparse ECM) or smaller (for a dense ECM) than the cell diameter.

## Cell cortex and nucleus

The cell cortex is represented as an elastic contour, with an elastic energy that penalizes stretching. At each reference arclength coordinate $s$ on the cortex, the net force density is a combination of elastic force and protrusive force,

$$\boldsymbol{F}^c(s) = \boldsymbol{F}^{c,\mathrm{el}}(s) + \boldsymbol{F}^{c,\mathrm{prot}}(s). \tag{2}$$

Elastic force densities on the cortex are defined as the derivative of a scalar tension times the tangent vector (in reference arclength coordinates). Let $\boldsymbol{\tau} = \partial \boldsymbol{X}/\partial s$ be the derivative of the cortex position with respect to reference arclength coordinates. Then the elastic force density is defined by [36]

$$\boldsymbol{F}^{c,\mathrm{el}}(s) = (T^c(\cdot)\boldsymbol{\tau}(\cdot))'(s). \tag{3}$$

Using the product rule, it can be seen that $\boldsymbol{F}^{c,\mathrm{el}}$ has both a normal and tangential component. $T^c(s)$ is a scalar tension on the cortex defined as

$$T^c(s) = k^c(s)(\|\boldsymbol{\tau}(s)\| - r(s)), \tag{4}$$

where the stiffness parameter $k^c(s)$ is analogous to a bulk modulus in 3D. To account for contractile forces, the stiffness $k^c$ used to compute the tension can vary in time and with arclength parameter $s$. Similarly, the rest length $r(s, t)$ of the tangent vector can vary in space and time. In the relaxed state, $r = 1$. When a region of the cell is contracting, we decrease $r$. Decreasing $r$

in a region of the cortex is equivalent to adding a positive active tension in that region. This kind of active tension is assumed in the form of $T^c$ used in [33, 34].

The cortex surrounds the nucleus, so that there is a small volume (area in 2D) of the cytoplasmic fluid between the nucleus and cortex. The cell nucleus is also modeled as an elastic contour. Since there are no protrusions, the total force density on the nucleus is given by the elastic force density

$$F^n = F^{n,\mathrm{el}} = (T^n \boldsymbol{\tau})'. \tag{5}$$

with $T^n = k^n(\|\boldsymbol{\tau}\| - 1)$. For the nucleus, the stiffness $k^n$ and rest length (set to 1) are constant throughout all simulations.

We next discuss the discretization of the forces. Defining some notation first, the cortex contour is discretized by $N^c$ initially equispaced nodes, while the nucleus contour is discretized at $N^n$ initially equispaced nodes. We denote the elastic force on the cortex at node $m$ by $F^{c,\mathrm{el}}_m$, where $m = 1, \ldots N^c$, and likewise for other quantities defined at the nodes. The discrete forces in Eqs (3) and (5) can then be computed using centered differences, with details given in the S1 Text, Section S2.

The final piece is the protrusive force density $F^{c,\mathrm{prot}}$. To define it, we first select a point $m^*$ on the cortex which is the "center" of the protrusion. Let the normal to the unit circle at point $m^*$ be denoted by $\boldsymbol{n}_{m^*}$. Then we define the protrusive force density by

$$F^{c,\mathrm{prot}}_m = \begin{cases} f_0 \boldsymbol{n}_{m^*} & m = m^* \\ \dfrac{1}{2} f_0 \boldsymbol{n}_{m^*} & |m - m^*| = 1 \\ -\dfrac{2f_0}{N^c - 3} \boldsymbol{n}_{m^*} & \text{otherwise.} \end{cases} \tag{6}$$

The protrusion force is therefore in the outward normal (at $m^*$) direction when $|m - m^*| \leq 1$, and the total force due to the protrusion is zero. The cell therefore does not move by exerting a force on the fluid, but the force on the nodes away from $m^*$ can result in the rear of the cell moving backward prior to the protrusion binding to the ECM, see, for example, Fig 2I(b) and 2II(b). The value for $f_0$ in Eq (6) is chosen to be above a certain threshold to overcome the elastic resistance of the cell. Our choice of $f_0$ results in a fast protrusion that does not contribute significantly to the dynamics of a single cell migration cycle, which is determined by viscous, elastic and contractile deformations.

Fig 2I(a) shows the spatial profile of the force density (in black) that generates the protrusion on the cortex (in green) in the 20 node ECM. Fig 2I(b) shows how the protrusion expands.

The physics of the continuum model are now completely defined. Given configurations of the cortex and nucleus, we compute the discrete force densities on the cortex (Eq (2)), and nucleus (Eq (5)). From these force densities, we multiply by the respective spacing between points in the reference configurations, $\Delta s = s_m - s_{m-1}$, to obtain forces, which we denote by $\hat{F}^c$ and $\hat{F}^n$. We add to these forces the spring forces due to the ECM (Eq (1)), and input these to the method of regularized Stokeslets. We then obtain the fluid velocity of the cortex, nucleus, and ECM points by regularizing the forces over a length scale of $\epsilon$ and solving the Stokes equations analytically. The points are then updated by a forward Euler method.

## Motility mechanisms

**Mechanism 1: Push-pull mesenchymal mode.**   In the first case, the cell randomly generates one protrusion at a time on its front half (facing the right). We do not investigate the process of choosing the direction of protrusion, as we focus on the mechanics of the motility cycle. Thus, we choose the node $m^*$ in Eq (6) on the right half of the cortex at random (i.e. from a uniform distribution). By specifying that the protrusion occurs in the front half of the cell, a preferred migration direction from the left to the right is specified in the model.

The subsequent cell movement pattern is shown in Fig 2I (top two rows). The cortex, which is initially relaxed (stiffness $k^c = k^{c,s}$) generates a protrusion (Fig 2I(a)) which extends until it comes into contact with an ECM node. Numerically, a contact occurs when the point at the tip of the protrusion comes within a distance $2\epsilon$ from an ECM node (twice the regularization length scale). The tip of the protrusion then reaches out and attaches to the ECM node, as shown in Fig 2I(b).

Once the cortex attaches to the ECM node, it immediately stiffens globally. Stiffening of the cortex is modeled by increasing the parameter for cortical tension $k^c$ to a value $k^{c,r}$, which is varied to investigate different motile regimes. The cortex then rapidly rounds to assume its circular resting configuration. In the case of a rigid ECM, the process results in the round cortex being pulled toward the attached ECM node. For the case of a softer ECM, the dynamics are more subtle. First, as shown in Fig 2I(c), the cortex assumes a quasi-circular shape by pulling the ECM node toward the cell. This inward pull continues until the elastic deformation force from the ECM balances the pulling force from cortical elasticity. The balancing of these opposing forces evolves, so that as the cortex becomes more circular, the ECM becomes less deformed, and the pulling force decreases. Eventually, the ECM node comes to rest near its initial position, as in Fig 2I(e). Once the velocities of the ECM and cortex nodes relax below a small threshold value, the cortex detaches from the ECM node. In order to physically uncouple the motion of the cortex and ECM node in the presence of the fluid, we move the ECM node a small threshold distance of $2\epsilon$ away from the cortex and wait for the system to equilibrate again (i.e., for the maximum velocity of the ECM and cortex nodes to relax below the small threshold value). The end result of this process is shown in Fig 2I(f). The cortex is then free to form another protrusion. We will refer to this entire process as the motility cycle, the steps of which can be summarized as follows.

1. Begin with $k^c = k^{c,s}$ and choose a random point $m^*$ on the $+x$ side (front half) of the cortex as the center of a protrusion.

2. Advance the system (by computing forces and velocities) until the tip of the protrusion reaches a distance $2\epsilon$ from the cortex node.

3. Manually set the location of the cortex protrusion tip to be equal to that of the ECM node ("bind" the protrusion). Globally increase the cortex stiffness to $k^c = k^{c,r}$

4. Advance the system (by computing forces and velocities) until the velocity of all points drops numerically below $\epsilon$.

5. Unbind the node by moving it a distance $2\epsilon$ away from the cortex.

6. Advance the system until the velocity of all points drops numerically below $\epsilon$, reset $k^c = k^{c,s}$, then go back to step 1 and repeat.

In step 2, it is possible that a protrusion could extend infinitely without contacting any nodes. For this reason, in mechanism 1 we automatically retract (by stiffening the cortex

globally) any protrusions that exceed a length of four cortical radii, assuming that they were not able to contact any nodes. We also refer to this case as a cycle, so that cycles can be either successful or unsuccessful. In fact, the case of unsuccessful cycles occurs frequently in real cells [37].

**Mechanism 2: Rear-squeezing mode.**    In this mechanism of motility, two protrusions are generated on the loose cell cortex. As in the push-pull mechanism, we choose the first protrusion location randomly on the front half of the cell. The second protrusion is generated at a location 15-45˚ apart from the first protrusion so that the protrusions are positioned to grow towards two adjacent ECM nodes. The resulting force distribution for these two protrusions is the superposition of two mechanism 1-type protrusions and is shown in Fig 2II(a). The protrusions expand until each contacts a node as in Fig 2II(b). Once contact occurs for both protrusions, the longer region of the cortex becomes very stiff, and we set $k^c = k^{c,r}$ and $r = 0.1$ in Eq (4) in this region only. We have used the same parameter $k^{c,r}$ across both mechanisms to represent increased cortical tension, although this tension is only increased in part of the cell in mechanism 2. Also, note again that decreasing the rest length to $r = 0.1$ is equivalent to adding an active tension (contraction) at the rear of the cell, and is required to generate enough tension to squeeze the cell through the ECM.

The shorter length, i.e., the leading edge at the right side, of the cortex between the two nodes remains relaxed (stiffness remains equal to $k^{c,s}$, and $r = 1$). The contracting rear of the cortex squeezes the nucleus through the gap between the two attached ECM nodes by deforming both the nucleus and ECM. This process results in cell propulsion, shown in Fig 2II(c)–2II(f). Once the system comes to a resting state (all node velocities decrease below a threshold), both nodes are moved a distance $2\epsilon$ away from the cell, and the system (cell and ECM nodes) is allowed to equilibrate until it comes to rest. This final state is shown in Fig 2II(g). The cell is then ready to form another protrusion. This entire process defines one cycle of motility, the steps of which can be summarized as follows.

1. Begin with $k^c(s) = k^{c,s}$ everywhere on the cortex and choose a random point $m_1^*$ on the $+x$ side (front half) of the cortex for the first protrusion.

2. Choose randomly a point $m_2^*$ which is $15 - 45˚$ away from $m_1^*$ to center the second protrusion.

3. Advance the system (by computing forces and velocities) until the tip of each protrusion comes within a distance $2\epsilon$ of an ECM node. The nodes must be distinct.

4. Manually set the locations of the cortex protrusion tips to be equal to those of their respective bound ECM nodes. Set $k^c = k^{c,s}$ and $r = 1$ in the shorter region of the cortex between the nodes. Set $k^c = k^{c,r}$ and $r = 0.1$ in the longer region of the cortex between the nodes.

5. Advance the system (by computing forces and velocities) until the velocity of all points drops numerically below $\epsilon$.

6. Unbind the nodes by moving each of them $2\epsilon$ away from the cortex. Reset $k^c = k^{c,s}$ and $r = 1$ everywhere.

7. Advance the system until the velocity of all points drops numerically below $\epsilon$, then go back to step 1 and repeat.

In step 3, two ECM contacts are required to move the cell, and protrusions sometimes fail to reach a node. As in mechanism 1, we retract protrusions that reach a certain length before hitting a node, and restart the process. In order to maintain the same total protrusive length across mechanisms, we stipulate the maximum length of each protrusion as two cortical radii,

so that the maximum total protrusive length of four cortical radii is the same as in mechanism 1. For the first cycle, we increase the threshold to three cortical radii to promote successful attachment of the initial protrusions. If only one of the protrusions contacts a node and the other reaches its maximum length without contact, the entire system is contracted (by increasing globally the cell stiffness) until the system returns to rest and can generate new protrusions. Although the cell does not move in this case, this too defines a cycle, so that once again cycles can be successful or unsuccessful.

## Parameter values and numerical procedure

**Physical parameters.**   Geometric parameters include the ECM mesh size and sizes of the cell cortex and nucleus, all of which are of the same order of magnitude [5, 38]. The cell diameter is often of the order 10 $\mu$m, while the nuclear diameter is sometimes only slightly smaller [17]. Thus, we choose the cell cortex and nucleus diameters at rest to be equal to 10 and 9 $\mu$m, respectively. We also consider the case in which the nuclear volume is significantly smaller than that of the cell. The length is normalized in all simulations to make the cell diameter, 10 $\mu$m, to be the unit of length. Openings in 3D extracellular environments range from 2 to 30 $\mu$m in diameter [39]. We vary the ECM density so that the average distance between the nearest ECM nodes is 1.5 or 0.5 length units in the cases of low- and high-density ECM, respectively. The cell can move through the ECM undeformed in the low-density case, and has to deform significantly to squeeze through the ECM in the high-density case.

The issue of physical dimensions in a 2D model that includes interactions of solid and fluid structures is subtle. From the hydrodynamic part of the model (see Section S1, S1 Text), it is clear that in the 2D method of regularized Stokeslets the force applied to the fluid at a point has the dimension of pN/$\mu$m (see Eqs. (S1.1) and (S1.3) in S1 Text; fluid velocity is in $\mu$m/sec, viscosity is in Pa·s, hence the force is in pN/$\mu$m). The interpretation is as follows: the 2D approximation is the planar cross-section of the 3D space, and the "per micron" factor appears in the force because this force is "per unit length in the direction perpendicular to the 2D plane." Simply speaking, to estimate the physical force in 3D space, one has to multiply the pN/$\mu$m force in the 2D model by the characteristic cell size, 10 $\mu$m. These considerations affect the choice of the mechanical model parameters.

There is, as expected, a significant variability in the reported mechanical characteristics of the ECM, nucleus, and cell cortex. In our 2D model, the ECM spring constant, $k^{\text{ECM}}$, is measured in units of pN/$\mu$m$^2$, so that when multiplied by the spring extension or shortening (in $\mu$m), the force on the point-like node has unit of pN/$\mu$m. The Young's modulus of the collagen mesh and ECM can vary from 1 to hundreds of Pa [12, 40–42]. We choose an ECM spring stiffness of $k^{\text{ECM}}$ = 50 pN/$\mu$m$^2$, which corresponds to the Young modulus within the range of values reported in the literature. This parameter is constant in all simulations.

The elastic and contractile force densities of the cell cortex and nucleus contours have dimensions pN/$\mu$m$^2$ in our 2D model, which, when multiplied by the characteristic distances $\Delta s$ between the discretization nodes in the contours, turn into forces with dimension pN/$\mu$m applied to the fluid. These force densities are spatial derivatives of the tensions along the contours, so these tensions have dimensions of pN/$\mu$m. Note that in Eq (3), the arc length $s$ is dimensional, in $\mu$m, while in Eq (4) the expression for strain in brackets is non-dimensional. The same considerations apply to the nuclear mechanics in the model. Thus, $k^c$ and $k^n$ are the dimensional proportionality coefficients in the cortex and nuclear contours' tensions, respectively, in pN/$\mu$m (the tensions are these proportionality coefficients times the non-dimensional strain). These proportionality coefficients are effectively spring constants for the respective

contours, which correspond to respective Young moduli (measured in $pN/\mu m^2$ in 3D) after being divided by the characteristic cell size, 10 $\mu$m.

The mechanical modulus of the nucleus can vary from tens of Pa [43] to hundreds of Pa [40, 44] to thousands of Pa [45], which corresponds to values of $k^n$ from units to hundreds of $pN/\mu m$. In the simulations, we vary the nuclear stiffness in a range even wider than that reported, from 1 to 10000 $pN/\mu m$, with 1—10 $pN/\mu m$ corresponding to 'soft', $\sim 100$ $pN/\mu m$ corresponding to baseline, and 1000—10000 $pN/\mu m$ corresponding to 'stiff' nucleus. In the tables below, we report the nuclear stiffness $\tilde{k}^n = k^n/10 \ \mu$m, to make consistent comparison with the ECM stiffness.

The tension of the cell cortex (in units of $pN/\mu m$, which directly corresponds to our parameters $k^{c,r}$ for a contracting cortex) was reported in the range from $\sim 100$ $pN/\mu m$ [46] to $\sim 1000$ $pN/\mu m$ [47]. We use the value $k^{c,r}$ varying up to 1000 $pN/\mu m$, except in one of the numerical experiments, where we simulate an exceedingly weak cortex with $k_c^r = 10$ $pN/\mu m$ (in which case, the cortex is still contracting due to a decrease in its tangent vector rest length $r$). For consistent comparison with the ECM stiffness, in the tables below we report $\tilde{k}^{c,r} = k^{c,r}/10 \ \mu$m. We use the value $k^{c,s} = 10$ $pN/\mu m$ for the loose, relaxed cortex, which corresponds to $\tilde{k}^{c,s} = 1$ $pN/\mu m^2$. Further discussion of the mechanics of nucleus and cortex can be found in [17, 48].

The last physical parameter we have in the model is the fluid viscosity. The actual physical value does not matter for the simulations, because the viscosity simply determines the time scale, even in the case of variable viscosities between the interstitial fluid, cytoplasm, and nucleoplasm. In the simulations, the viscosity is normalized to 1 and does not vary across the interstitial fluid, cytoplasm, and nucleoplasm. This is for simplicity of numerical computation, as variable viscosities complicate the equations and numerical simulations. Nevertheless, it is instructive to discuss the physical values of the viscosity of the nucleoplasm, cytoplasm and interstitial fluid which could be one to three, and even four orders of magnitude greater than the viscosity of water [41, 42, 49–51]. In physical units, this means viscosities from 0.01 to 10 Pa·s. In the process of 3D cell migration, the timescale of viscous relaxation was observed to be on the order of tens of seconds [44]. In our simulations, characteristic net tensions of $\sim 10$ $pN/\mu m$ (greater elastic and contractile forces largely equilibrate, so the net force is relatively small), characteristic sizes of 10 $\mu$m and characteristic viscosities of 10 Pa·s result in a $\sim 10$ s time scale which compares well with observations reported in [44]. Of course, the additional processes of developing and relaxing protrusions, adhering to the ECM, and contracting can add a significant time to this estimate; no wonder that 3D motile cycles were reported to take tens of minutes [52].

**Numerical parameters.** The number of grid points for both the cortex, $N^c$, and nucleus, $N^n$, contours in the simulations were varied depending on the motility mechanism. In general, we set $N^c = N^n = 80$ for mechanism 1 and $N^c = 120$, $N^n = 80$ for mechanism 2. The number of cortex points is larger for mechanism 2 because the cell gets closer to the ECM nodes, which can leak inside the cell if the cortical boundary is insufficiently resolved. While the fluid technically prevents this from happening, there is still discretization error in our model. Since the discretization of each contour has a finite number of points, discrete ECM and nucleus points could pass through regions between the cortex nodes and ruin the geometry of our simulations. We remedy this in two ways. First, if the nucleus in the model has zero physical bending rigidity, we add a small computational bending rigidity with an equispaced reference configuration as described in the S1 Text, Section S2. We do this only when the protrusions are forming, and not when the cell is in a contractile state, so that the computational bending forces prevent the nucleus from being squeezed into thin protrusions. Second, we check, at every timestep, whether an ECM node is inside the polygon that defines the discrete cortex. If there

is such a node, we move it a distance $\epsilon$ in a random direction until it is outside the polygon defining the cortex. We do the same for cortex nodes inside the nucleus, and then proceed to the next timestep.

The maximum stable time step is dictated by ECM stiffness and is 0.001 (in dimensionless time units) for $k^{\text{ECM}} = 50$ pN/$\mu$m$^2$. We decrease this timestep down to 0.0002 for a short period of time ($\mathcal{O}(0.1)$ time units) when the cortex stiffens as it binds to the ECM nodes.

Mechanism 2 is much more challenging numerically than mechanism 1 because of the changing spring rest length $r$ in Eq (4). Once the cortex binds to the ECM nodes, grid points behind the ECM nodes begin to pack very close together due to the shorter rest length. Meanwhile, points that span the front section of the cortex between the two ECM nodes grow apart quickly as the entire nucleus squeezes through the two nodes. The region in front of the two nodes therefore becomes under-resolved, and dynamic remeshing is required to keep a stable configuration of the cortex and nucleus. This remeshing takes place every $\mathcal{O}(100)$ time steps and only when the cell is attached to two ECM nodes. Because the nucleus interacts with the cortex via hydrodynamic forces, it too becomes under-resolved at the front, so the remeshing is applied to both the cortex and the nucleus.

Our goal is to dynamically update the discretization so that the spacing between the grid points stays roughly constant through time. We accomplish this as follows. Given a current set of the grid points on the cortex/nucleus $X_j$ with reference points $X_j^0$, we use the positions $X_j$ to construct a continuous cumulative arclength function, defined as $s(j)$, where $j$ is the point index (for non-integer values of $j$, we compute $s(j)$ by linear interpolation between the two closest integer values). We then sample $s(j)$ at equal lengths in $s$. Each of these samples corresponds to a value of $j$. We denote the set of equally spaced arc lengths as $S(j^{\text{eq}})$. Then the new point positions are given by $X(S(j^{\text{eq}}))$ and the corresponding new reference points are given by $X^0(S(j^{\text{eq}}))$. Since $X$ is only known at indices $X_j$, we use linear interpolation to obtain $X(S(j^{\text{eq}}))$ if $j^{\text{eq}}$ is not an integer. We then update the positions of the grid points along with their reference configurations according to the new positions $X(S(j^{\text{eq}}))$ and $X^0(S(j^{\text{eq}}))$. The reference configurations are then used to compute elastic and bending forces as described in Section S2, S1 Text. At the start of each mechanism 2 cycle, the cortex reference configuration is reset to be equispaced so that each cycle begins similarly. See Section S3 of S1 Text for more details about the remeshing algorithm.

## Results

### Cells can use both mechanisms to move through a sparse ECM

Simulations of a cell moving through a sparse ECM (20 nodes; the mesh size is greater than the cell size) over one motility cycle are shown in Fig 2. Additional time values are provided in S2 Fig for a cell migrating with the push-pull mechanism and in S1 Video. A cell migrating with a rear-squeezing mechanism in a sparse ECM is shown in S3 Video. The figures and movies show that the cell is able to move through the ECM with minor deformation. In this regime, the limiting factor is not mechanics, but rather the timing of developing protrusions and attachments to the ECM. In particular, simulation data show that the cell can move without deforming its nucleus or the ECM. It can therefore move effectively regardless of the nuclear stiffness and contractile force.

Experimentally, it was observed that, in sparse matrices, cells use mechanisms entirely different from the ones we are considering. These include wrapping tightly around one ECM fiber and moving along it [4, 53], which intuitively is a much more logical way to move than trying to search for far away ECM nodes. Consequently, we do not investigate the sparse ECM further and turn to the case of a dense ECM of 60 nodes (see Fig 1(b)).

## Mechanism 1 performs best for high tension and soft nucleus

We next consider a denser ECM (60 nodes; the ECM mesh size is roughly half the cell size) and a few characteristic regimes of the parameters and analyze the cellular motion therein. There are three characteristic force scales in the process: (1) $T$, the characteristic contractile tension in the cortex, the order of magnitude of which is given by the parameter $k^{c,r}$. (2) $E$, the characteristic tension of the ECM deformed by the nucleus squeezing through it. This tension is of the order of the ECM spring coefficient $k^{ECM}$ multiplied by the cell size minus the average ECM mesh size. (3) $N$, the characteristic tension in the deformed nuclear envelope, which is of the order of the nuclear stiffness $k^n$. In this section we consider all qualitatively different relations between these three forces.

Table 1 gives the parameter choices for each of these regimes. We note that the ratio of the forces with respect to one another, rather than their actual physical values, is used to characterize different parameter regimes. The actual value of the model parameters for $E$, $T$, and $N$ to simulate a particular motility regime may depend on other model parameters. For example, in simulations where the diameter of the nucleus is halved, we show that the regime where nuclear force is dominant, $N > T > E$, cannot be simulated in the model (since the nucleus is too small to be the stiffest component).

We begin by discussing a regime in which the cell cannot move. This regime is actually composed of two orderings of the forces: $E > N > T$ and $N > E > T$. In either case, both the force required to deform the nucleus and the force required to deform the ECM are greater than the contractile force. In this case, the position of the cortex, nucleus, and ECM at two time values are shown in Fig 3I(a). Additional simulation time values are shown in S3 Fig and in S2 Video. Results show that the cell becomes "stuck" in the ECM and is unable to pass through the small gaps in the dense ECM.

We next consider the case $T > E > N$, when the contractile force is the largest and the nuclear force is the weakest. The positions of the cell and ECM from a simulation are shown in Fig 3I(b) with more time values provided in S4 Fig. Data show that the nucleus is highly deformable and can easily slide through the gaps between the ECM nodes for these parameter values. The strong contractile force contracts the cell rear, so that the entire cell is able to slide through the ECM efficiently.

When the force required to deform the nucleus increases, the cell moves by deforming the ECM. The cell and ECM positions from simulations in the parameter regime $T > N > E$ are shown in Fig 3I(c) and S5 Fig. The difference between the regimes $T > N > E$ and $T > E > N$ can be seen in Fig 3 (compare I(c) to I(b)), where the ECM nodes are further displaced from their original locations (shown as black ×'s) in the former regime than the latter. In the

**Table 1. Parameter values for the different forcing regimes for both migration mechanisms.**

| Push-pull mechanism | | | Rear squeeze mechanism | | |
|---|---|---|---|---|---|
| Regime | $\tilde{k}^n$ (pN/μm²) | $\tilde{k}^{c,r}$ (pN/μm²) | Regime | $\tilde{k}^n$ (pN/μm²) | $\tilde{k}^{c,r}$ (pN/μm²) |
| $E > T, N > T$ | 100 | 1 | $E > T, N > T$ | 10 | 1 |
| $T > E > N$ | 1 | 100 | $T > E > N$ | 0.1 | 100 |
| $T > N > E$ | 10 | 100 | $T > N > E$ | 10 | 100 |
| $N > T > E$ | 100 | 50 | $N > T > E$ | 1000 | 100 |
| $E > T > N$ | 1 | 1 | $E > T > N$ | 0.1 | 10 |

Additional simulation information: there are 60 ECM nodes and $k^{ECM}$ = 50 pN/μm².

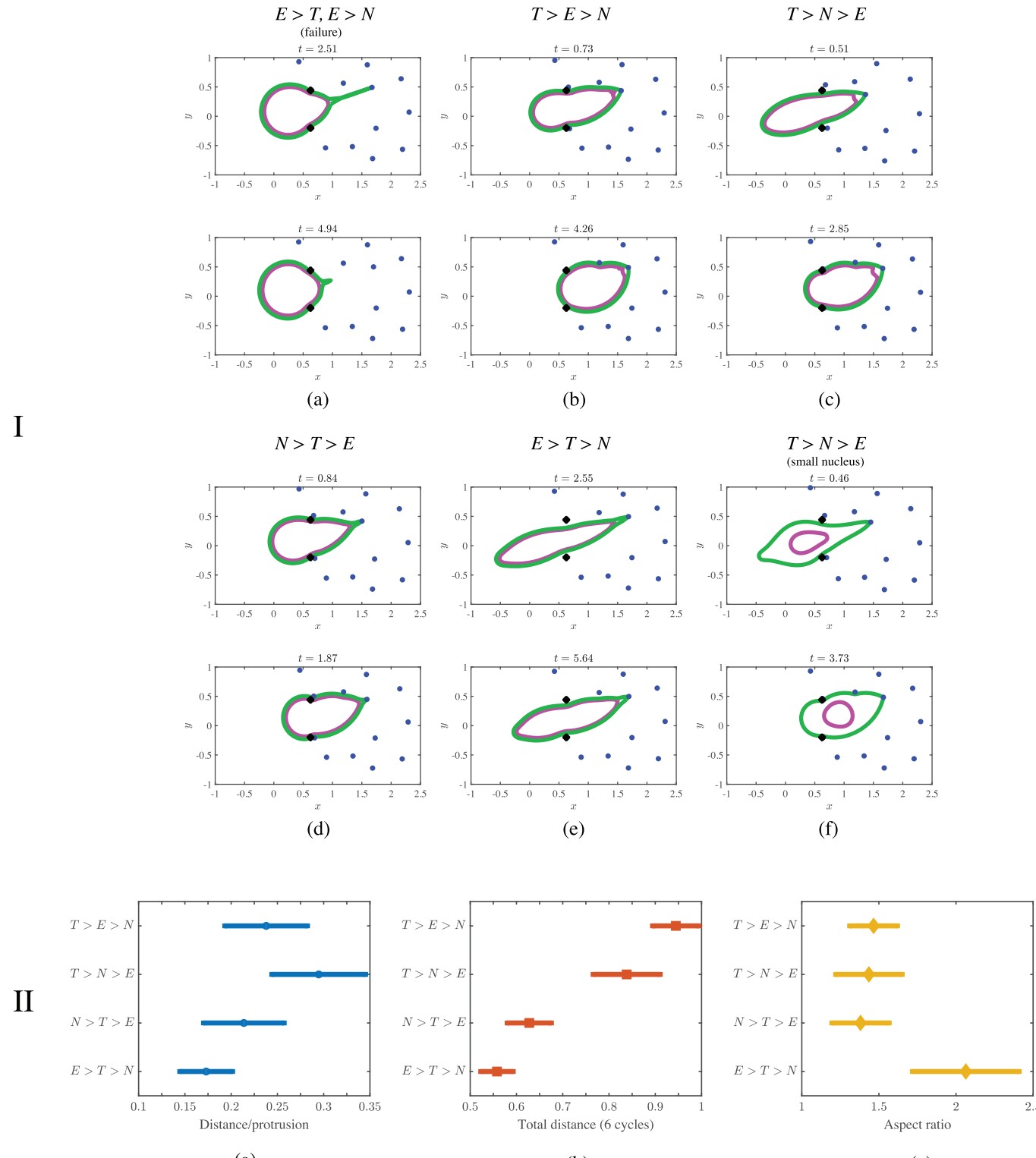

**Fig 3. Results from simulating cell migration using the push-pull mechanism.** Panel I shows the position of the cortex (green), nucleus (magenta), ECM nodes (blue), and initial position of the ECM (black ×'s) over a range of parameter values listed in Table 1. Simulation time values are located above each panel. The bottom panel II shows the normalized distance of the cell from its initial location (distance is normalized by the cell diameter at rest). The average distance traveled over one cycle is shown in (a). The average distance per simulation (total displacement over 6 cycles) for each of the parameter regimes simulated is shown in (b). The data show a stronger contractile force T leads to maximum displacement for both measurements. (c) The mean aspect ratio over time, defined

in Eq (7), provides a measurement of how elongated the cortex becomes during a simulation. Data show that the regime $E > T > N$ is characterized by the longest, thinnest cell. Error bars are a single standard error in the mean in each direction (70% confidence intervals).

$T > N > E$ regime, the net displacement of the nucleus is reduced by about 10% compared to the regime $T > E > N$ (see S2 Table).

Similarly, we observe displacement of the ECM in the parameter regime $N > T > E$, when the nuclear stiffness is relatively largest. In this case, the cell has the roundest shape of all parameter regimes simulated, and the cell migrates by deforming the ECM, as shown in S6 Fig. In fact, the maximum ECM displacement for the data in S6 Fig is approximately 60% larger than that for the parameter regime $T > N > E$ (data shown in S5 Fig). The net displacement of the nucleus in Fig 3I(d) is also reduced by about 30% compared to Fig 3I(c) (see S2 Table). Overall, the relatively large stiffness of the nucleus causes the cell to maintain a round shape during migration. Therefore, the ECM must deform more than in other parameter regimes for successful motility, and hence the effectiveness of motion in this regime is reduced.

For a cell with a smaller nucleus, the parameter regime $N > T > E$ cannot be simulated, even with a stiff nucleus, because the nucleus is not large enough to limit motility at the simulated ECM density. To show this, we simulate again with the $N > T > E$ parameters in Table 1, but with the nuclear diameter reduced by half. As shown in Fig 3I(f) and S8 Fig, simulation data on cell shape and distance traveled are consistent with simulations with a larger nucleus in the $T > N > E$ parameter regime. Therefore, we label the small nucleus simulation as $T > N > E$ since cortical tension contributes most significantly to migration in this case.

Finally, we consider the case when the nucleus is easy to deform, but there is not enough tension to deform the ECM. This is the parameter regime $E > T > N$, and under these parameters simulation data in Fig 3I(e) show essentially rigid ECM nodes that do not deviate from their original positions. The cell must therefore move by deforming its nucleus. Position data in Fig 3I(e) (additional time values for the positions of the cortex, nucleus, and ECM nodes are provided in the S7 Fig) show that the nucleus is able to slide through the gaps by assuming a long and thin shape, but the relatively weaker contractile force is sometimes unable to efficiently contract the rear of the cell. Several motility cycles are necessary for the entire cell to squeeze through a gap in the ECM.

**Quantitative comparison of parameter regimes.**   To generalize our results to random migrations, we performed 8 simulations for the parameter regimes listed in Table 1, where the cell undergoes 6 protrusion cycles (protrude, bind, release, relax, repeat) per simulation (the total number of protrusions is 48). We measure the net distance moved per cycle and the net distance moved at the end of the simulation (total displacement) for the different parameter regimes. Distances are measured by tracking the center of mass of the nucleus (mean coordinates of the discrete points around the nuclear contour). To compare the morphology of the cells, we also measure the mean aspect ratio of the cortex in each regime. We define

$$a = \frac{x_{\max} - x_{\min}}{y_{\max} - y_{\min}}, \tag{7}$$

as the discrete aspect ratio of the cell, where the maximum and minimum are taken over the discrete points of the cortex contour. In Fig 3II(c), we show the mean and standard deviation over time of $\max(a, 1/a)$ (this last maximum is used so that the aspect ratio is always larger than 1).

Quantitative data in Fig 3II show the optimal parameter regimes for cell migration are $T > E > N$ and $T > N > E$. The total distance the cell moves averaged over six cycles (Fig 3II(b)) in

these regimes is larger than for other simulated parameter regimes. The data for an average over one cycle do not conclusively separate the regimes (Fig 3II(a)). Results from our simulations show that cells with large enough cortical tension are able to overcome the characteristic force required to deform the nucleus and/or ECM.

Less effective parameter regimes are characterized by relatively reduced cortical tension. Between the low-tension regimes $N > T > E$ and $E > T > N$, the regime with a stiffer nucleus appears to perform slightly better. One reason for this might be the cell's large aspect ratio when $E > T > N$. As shown in Fig 3II(c), the parameter regime $E > T > N$ is characterized by the largest aspect ratio, meaning that the cell is long and thin when migrating. Since the front of the cell is always bound to an ECM node, larger aspect ratios imply that the cell's center of mass travels a smaller distance than for other parameter regimes.

To summarize the results from the first mechanism, simulations of cell migration using parameter regimes with high tension are able to migrate most effectively, regardless of the force required to deform the ECM. When the nuclear force or ECM force exceeds the tension on the cortex, the cell can still migrate, but it covers substantially less distance.

## Mechanism 2 results in robust cell migration

We analyze five different parameter regimes representing combinations of three characteristic forces for cells migrating using mechanism 2. Parameter values and combinations are listed in Table 1. We again point out that the actual parameter values of $E$, $T$, and $N$ that characterize each regime in Table 1 may change with other model parameters. For example, when the diameter of the nucleus is halved, the nuclear force $N$ becomes much smaller (see S15 Fig). Likewise, when the nucleus is given a finite bending rigidity, the nuclear force $N$ increases (see S16 Fig).

Simulation results show that for a sparse ECM, a cell is able to successfully migrate as long as it can find two nodes to bind to (see S3 Video). The positions of the membrane, cortex, and ECM during a simulation of one motility cycle in this case are shown in Fig 2II.

For a dense ECM, the cell is unable to migrate in the parameter regime $E > T$ and $N > T$. The positions of the cortex, nucleus, and ECM at two time values from a simulation are shown in Fig 4I(a) with several more time values provided in S10 Fig. A movie showing the position of the cell and ECM during a simulation is provided in S4 Video. The data show that when the cortex contraction is too weak to deform either the nucleus or ECM, the cell is not able to squeeze through the ECM gap.

When the contractile tension on the cortex is increased, for example in the parameter regime $T > E > N$, the cell is easily able to pass through the ECM network. In this case the cell and ECM position during migration are graphed in Fig 4I(b) with more time values provided in S11 Fig. The cortex contracts at the rear and squeezes the nucleus through the gap between ECM nodes.

When the nucleus is relatively stiffer in the parameter regime $T > N > E$, the cell does not completely pass through the ECM after one motility cycle (compare Fig 4I(c) to 4I(b)). The nucleus in Fig 4I(c) (top panel) appears to "buckle" under high cortical tension (more time values from a simulation in the parameter regime $T > N > E$ are shown in S12 Fig and S5 Video). This buckling, which is an artifact of the stretching energy Eq (3) penalizing stretching but not shear or bending, may be biologically irrelevant because such deformations of the nucleus could be prevented by a structure such as the perinuclear actin cap [54]. That said, S4 Table shows the difference in the nucleus' center of mass is $< 1\%$ between Fig 4I(b) and 4I(c). Thus our simulations show that mechanism 2 is approximately equally effective if $T > E$, regardless of nuclear stiffness.

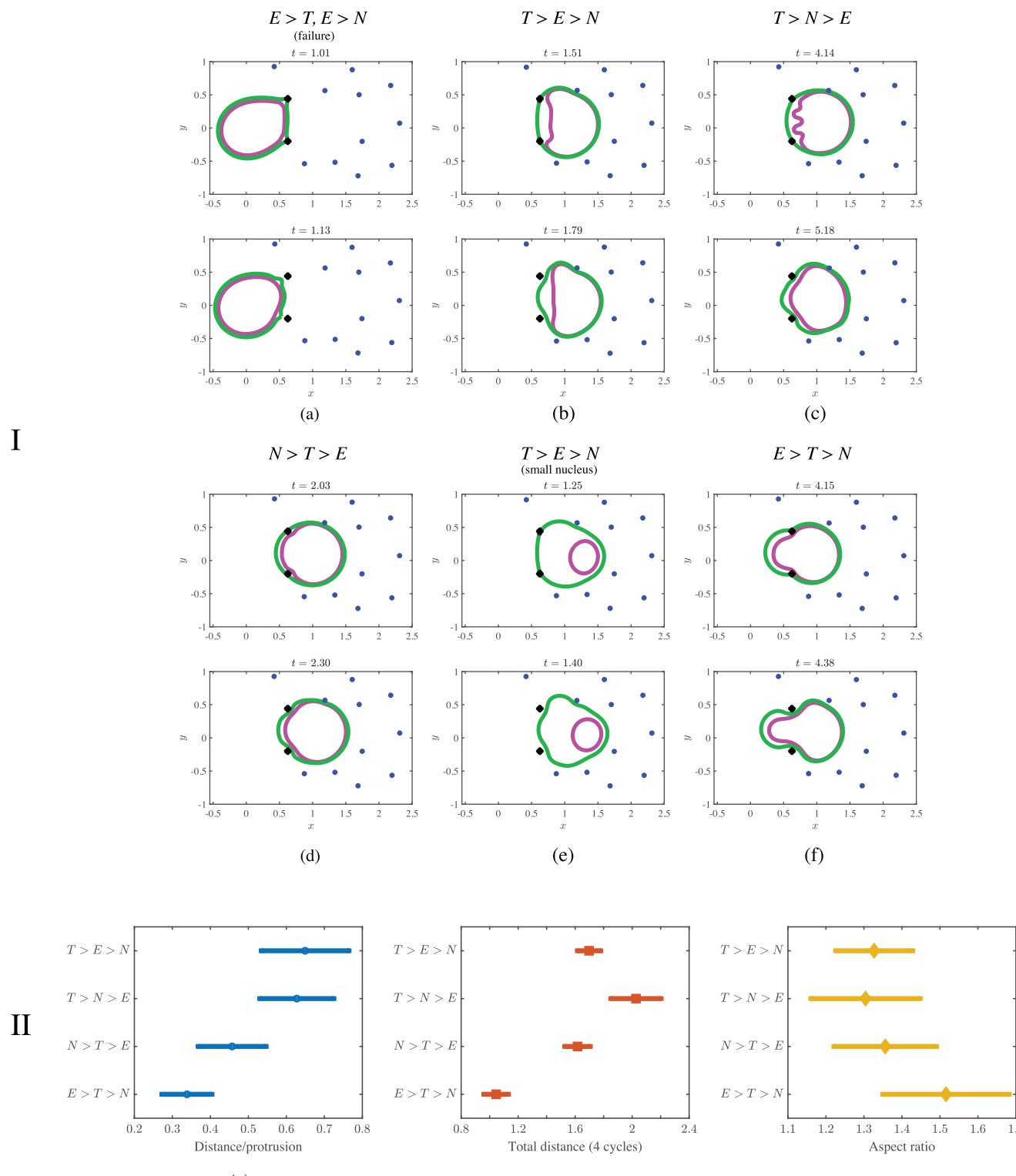

**Fig 4. Results from simulating cell migration using the rear squeezing mechanism.** Panel I shows the position of the cortex (green), nucleus (magenta), ECM nodes (blue), and initial position of the ECM (black ×'s) over a range of parameter values listed in Table 1. Simulation time values are located above each panel. Panel II shows the normalized distance traveled by the cell from its initial location (distance is normalized by the cell diameter at rest). The average distance moved over one cycle is shown in (a). The average distance per simulation (total displacement over 4 cycles) for each of the parameter regimes simulated is shown in (b). (c) The mean aspect ratio over time, defined in Eq (7), provides a measurement of how elongated the cortex becomes

during a simulation. Data show that the regime $E > T > N$ is characterized by the longest, thinnest cell. Error bars are a single standard error in the mean in each direction (70% confidence intervals).

The cell also migrates effectively in the parameter regime $N > T > E$, where the nuclear stiffness is large relative to other parameters, but the cell has enough tension to deform the ECM. Fig 4I(d) shows an overall round cell during a successful motility cycle (additional simulation time values are provided in S13 Fig). The cell migrates by deforming the relatively soft ECM nodes. Note the increased stiffness of the nucleus prevents the buckling seen in Fig 4I(c). While the cell's passage in Fig 4I(d) through the ECM gap is not complete, S4 Table shows the decrease in the mean nuclear displacement is only about 5% compared to the regime $T > E > N$. Our results show that the cell migrates robustly with respect to nuclear stiffness for simulations in the parameter regimes $T > E$. For this reason we refer to mechanism 2 as *more robust* to changes in parameters than mechanism 1.

As in mechanism 1, the parameter regime $N > T > E$ is defined in part by the large size of the nucleus, since simulations with a smaller nucleus do not fall into this regime regardless of nuclear stiffness. To show this, we use the $N > T > E$ parameters in Table 1 and reduce the nuclear diameter by half to simulate a stiff, small nucleus. As shown in Fig 4II(e) and S15 Fig, simulation data on cell shape and distance traveled are consistent with simulations with a larger nucleus in the regime of weakest nuclear force, $T > E > N$. Therefore, we label the small nucleus simulation as $T > E > N$ since cortical tension contributes most significantly to migration in this case.

When the tension on the cortex $T$ is less than the required force to deform the ECM $E$, the cell's migration is reduced and the cell's shape is characterized by a larger aspect ratio, similar to results from mechanism 1. Fig 4I(f) shows a highly deformed nucleus as the cell migrates through a gap (additional time values graphed in S14 Fig). While migration is successful, the cortex and nucleus do not completely clear the gap after one motility cycle because contraction on the cortex is not large enough relative to ECM stiffness. In this case, the nucleus' center of mass travels about 80% of the distance it travels when $T > E > N$ (see S4 Table).

**Quantitative comparison of parameter regimes.** As for mechanism 1, we perform 8 simulations, where for each simulation the cell goes through 4 cycles of mechanism 2 (we perform 4 cycles because the distance per cycle is larger than in mechanism 1). We again measure the net distance migrated per cycle and the net distance moved (total displacement) at the end of the simulation across the different parameter regimes listed in Table 1. There are a total of 32 total cycles analyzed. Results are shown in Fig 4II.

Results from simulation data show that mechanism 2 is robust to changes in nuclear stiffness. The total distance per protrusion (Fig 4II(a)) and total distance per cycle (Fig 4II(b)) are the same within error for the three parameter regimes with $T > E$. Similar to results from mechanism 1, there is a reduction in motility when $E > T$ as the cell becomes more elongated and more cycles are required to achieve the same amount of motion (see aspect ratios in Fig 4II(c)).

Comparing quantitative results from simulations where the cell uses mechanism 1 (push-pull) to mechanism 2 (rear-squeezing), mechanism 2 results in relatively increased distance traveled over a range of parameter values. Although the total distance traveled in the regime $E > T > N$ is approximately 60% of the distance traveled when $T > E > N$ in both mechanisms, mechanism 2 performs much more robustly when the nuclear stiffness changes. In particular, mechanism 2 shows no significant drop in the total distance moved as nuclear stiffness changes, as long as $T > E$ (Fig 4II(b)). This is in contrast to mechanism 1, which shows the distance traveled dropping significantly as the force to deform the nucleus $N$ increases

(Fig 3II(b)). For this reason, we describe mechanism 2 as more robust than mechanism 1. Furthermore, in all regimes, the displacements of the cell are greater for mechanism 2 than for mechanism 1. That said, we note that a cell migrating using mechanism 2 experiences increased cortical tension compared to mechanism 1 because the rest length of the cortex at the back of the cell in this mechanism is decreased by 90%.

### Mechanism 2 is aided by hydrodynamics

Pressure, velocity, and speed during a motility cycle for both mechanisms of motility during comparable stages are shown in Fig 5 for the parameter regimes $E, N > T$ (Fig 5(a)) and $T > E > N$ (Fig 5(b)) listed in Table 1. When cortical tension is relatively small, the cell fails to migrate. Regions of high pressure develop near the ECM nodes (Fig 5I and 5II(a)), the cell is sterically inhibited from migrating through the nodes, and the speed of the fluid is relatively small.

Data in Fig 5I and 5II(b) show the case when cortical tension is the dominant force, and the cell migrates effectively. The cell in Fig 5I(b) is in the 'ECM pull' stage of mechanism 1 (see Fig 2I(d)), and the cell in Fig 5II(b) is in the 'squeezing through nodes' stage of mechanism 2 shown in Fig 2II(d). When there is a favorable pressure gradient, fluid flows from regions of high to low pressure, but fluid flow can still occur from low to high pressure. One example occurs in flow past an airfoil [55]. Fig 5II(b) shows there is a favorable pressure gradient with high pressure in the cell rear along with low pressure in the front of the cell for mechanism 2. In mechanism 2, the active tension on the rear of the cell membrane results in a pressure jump, as shown in Fig 5II(b). No pressure jump is observed at the front of the membrane because of reduced tension at the leading edge of the cell. The pressure gradient, which results from compression of the fluid at the rear of the cortex in mechanism 2, induces an additional fluid velocity in the direction of migration. Thus, hydrodynamics and incompressibility of the fluid *aid* migration in mechanism 2.

The opposite scenario is observed for the push-pull mechanism of motility, where the pressure is low in the cell rear and high at the front of the cell near the protrusion (see Fig 5I). The fluid is compressed at the front of the cell due to the cortical deformation at the leading protrusion as well as from the deformation of the ECM node. In spite of this adverse pressure gradient, the horizontal component of the cell's velocity (as well as that of the fluid) is positive (Fig 5(a), bottom) so that the cell is migrating from left to right. The Stokes equations (see Eq. (S1.1) in Section S1 in S1 Text) are a force balance so that the restoring force of the ECM overcomes the pressure gradient force, enabling the cell to migrate. A simple analogy is to consider a rubber band that is stretched in a direction parallel to a pressure-driven background flow. Once the rubber band is sufficiently stretched and released, it will move in a direction opposite the fluid flow (from low to high pressure), dragging fluid with it. Fig 5 shows that the ECM acts as this kind of rubber band in the push-pull mechanism of motility.

## Discussion

In this study, we have considered two mechanisms of cell migration: a mesenchymal-like push-pull and an amoeboid-like rear-squeezing mechanism. The three primary mechanical parameters of the model are the elasticities of the ECM and nucleus and the contractile tension of the cortex. The elasticity of the ECM determines the characteristic force required to stretch the ECM gap to the size of the undeformed nucleus, and the nuclear stiffness determines the force required to squeeze the nucleus to the width of the undeformed ECM gap. We find that the relation between three forces—the cortex's contractile force and the characteristic forces of the ECM and nucleus deformations—determine the effectiveness of cell migration for densely

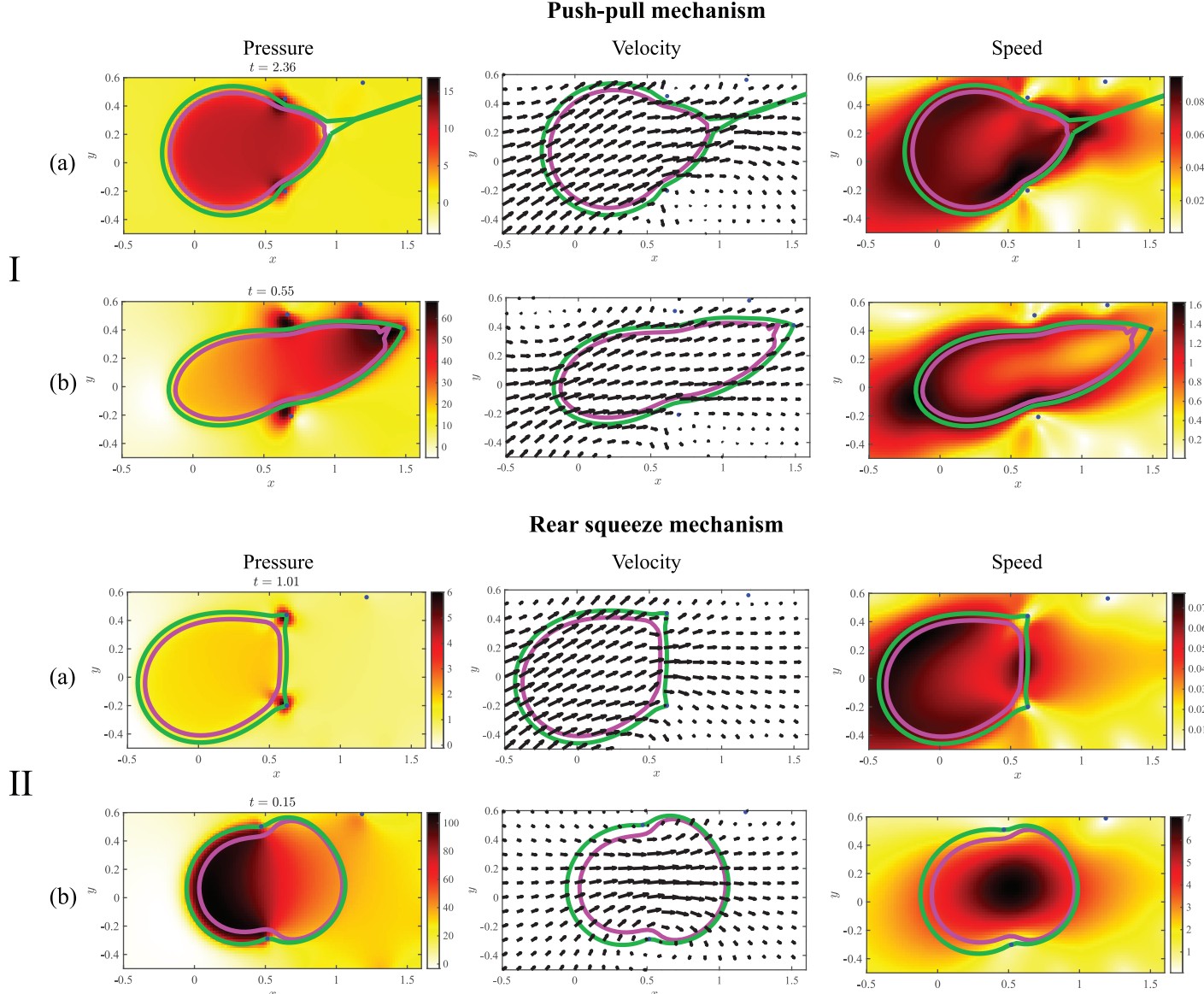

**Fig 5. Pressure, velocity, and speed for a for a cell using the push-pull (I) and rear-squeezing (II) mechanisms of motility.** The position of the cortex (green), nucleus (magenta), and ECM nodes (blue) are also shown. Note the velocity vectors are scaled to show the direction of the flow, while the speed measures the magnitude of the velocity vectors. Data in (a) correspond to simulation parameters $E, N > T$ in Table 1 when the cell cannot move. Data in (b) are generated using simulation parameters corresponding to $T > E > N$ in Table 1.

packed ECMs where the gap size is smaller than the nuclear diameter. The cell migrates most effectively when the contractile force exceeds both characteristic deformation forces, but the cell can also migrate persistently even if its tension is too weak to deform a near-rigid ECM, as long as the cortex's contraction is sufficient to deform the nucleus. The cell can also migrate even if it fails to deform a stiff nucleus, but then the cell has to be able to sufficiently deform the ECM. Thus, the intuitive conclusion is that the cell has to contract enough to deform either the ECM or the nucleus, but not necessarily both. While the cell's motion is reduced when contraction is reduced, a non-trivial conclusion of our study is that the regime when the cell can deform the nucleus but not the ECM is least effective for migration. We showed that these

deformations result in elongated cells with high aspect ratios, and that several cycles of protrusions are necessary to move the rear of the cell through the ECM. Thus we showed that large aspect ratios lead to smaller distances traveled.

Another nontrivial prediction is that the rear-squeezing motility performs better than the push-pull mechanism in that the distance traveled per cycle is larger and experiences less variation when the nuclear stiffness changes. One reason for this prediction is the geometry of the model: in the push-pull mode, it is more likely, but not necessarily given, that the protrusion is made to the nearest node in front of the cell. The cell is then pulled to that node, and in a dense ECM network, the distance between the cell and the node is smaller than the cell diameter. In the rear-squeezing mode, the cell is squeezed forward in between and through two nodes that are closer originally to the cell front, and so the cell often advances almost one body length. This prediction has to be used with caution: in reality, push-pull protrusions could be made to far-away nodes, in which case the push-pull mode would become more effective. Conversely, the rear-squeezing mechanism could be less effective if two random attachments to ECM nodes were used. Suppose in particular that the cell attempted to use two non-adjacent nodes to pass through a "gap" containing another node. In this case, the elasticity of the center node would block the cell from moving, since at some point the elastic force on the center node will push against the motion of the cell. In this study, we tried to exclude these cases by mandating that protrusions be 15-45° apart on the cortex contour.

Another relevant note is that we only investigated the mechanical constraints for the migration, not kinetic ones, as we did not examine how the speed of migration varies with changing parameters. In this setting a sparse ECM would fare poorly since the cell migration cycles would be prolonged by making protrusions that fail to bind to any nodes. In fact, it was observed that when a cell encounters just one fiber, the cell adopts a spindle-like morphology and migrates effectively in 1D along this fiber [53, 56]. Our conclusion that the rear-squeezing motility performs better than the push-pull mechanism is also limited to the mechanical performance of the cell, and does not take into account the time needed to extend protrusions in the right directions and establish adhesions with the ECM fibers. For the rear-squeezing motility mechanism, more than one such successful protrusion/adhesion processes have to occur, compared to just one in the push-pull mechanism, and it is entirely possible that these processes take much longer time in rear-squeezing motility. In addition, differences between efficiencies of signaling processes regulating the protrusion/adhesion activity at the cell rear versus front are unknown.

An important feature of our model is the underlying fluid and hydrodynamic forces. The hydrodynamic forces and movements effectively determine the interaction between the nuclear and cortical boundaries, as well as the interaction of each of these with the ECM. For example, in the rear-squeezing mode, a higher tension is required because the contraction at the rear must overcome the elasticity of the nucleus, which resists the contraction of the cortex through the hydrodynamic forces. Furthermore, because of the hydrodynamic forces, when the cell is stuck in the ECM, it truly is stuck, as any motion of the cell must carry the ECM with it due to the underlying fluid. The cell cannot slip away from the ECM unless there is sufficient space between it and the nodes. In this sense, we effectively model the parameter regimes where the cell is sufficiently loose to stay away from the ECM. Also, many previous studies have used artificial viscous dashpots to account for the movements and deformations generated by the elastic and contractile forces. In our case, the movements and deformations are mediated by actual viscous fluid. As a consequence, the presence of the fluid determines a timescale of the migration via a physical parameter, the viscosity, which can be measured experimentally. Lastly, our approach allows simulating the fluid pressure, the significance of which is highlighted by the prediction of the importance of nucleoplasmic pressure in pushing the nucleus through the ECM [40]. In our study, we determined that a favorable pressure

gradient aids migration for cells using a rear-squeezing mechanism. We showed that intracellular pressure is high in the cell rear and low in the cell front so that the hydrodynamic contribution from fluid incompressibility aids migration. In contrast, we observe an adverse pressure gradient for the push-pull mechanism of migration.

The conclusions of our model are in qualitative agreement with a number of experimental studies. For example, dendritic cells of the immune system have soft and deformable nuclei, allowing them to move at a rapid speed of a few microns per minute in 3D [16]. By contrast, fibroblasts have stiffer nuclei and move through the ECM at a rate slower than one micron per minute [4]. Cells reportedly move faster in softer ECM [12]. Cells were observed to move slower or even get stuck in conditions with stiffer nucleus or smaller ECM mesh size [57]. In fact, the problem of nuclear deformation is so dramatic that cells are known to use actomyosin cytoskeletal networks to actively deform the nucleus when migrating through narrow gaps [58]. Cells even use the elastic nucleus as a gauge to measure pore sizes and go to the least resistance path [59]. Last, but not least, one report supports the predicted advantage of the rear-squeezing mechanism: when neutrophils migrated through small holes, actin was seen to concentrate mostly at the cell rear, and myosin contraction was necessary to propel the cells through the holes [58].

Our simple model has, of course, a number of limitations. The main one is that we investigated a 2D caricature of the 3D geometry; the full 3D simulation would be prohibitively computationally expensive for multiple parameter sweeps. The structure of the cell in the model is simplified; for example, the model does not include mechanical connections between the lamin-based nucleoskeleton and the cytoskeleton, which facilitate force transmission between the nucleus and the extracellular matrix [60]. We have not explored in detail dependence of the motile behavior on mesh size and adhesion characteristics. The ECM, cortex and nucleus in our model are a linear elastic network, while the mechanical properties of actual ECM, cortex and nucleus are very complex [38, 61]. Needless to say, we have not investigated how cells change migration mode in response to the ECM properties [62], and we have not considered other modes of motility, such as blebbing [34] and chimneying [63]. More advanced problems, like mechanosensing [64], ECM remodeling and its feedback with cell mechanics [65], coordination of traction forces with ECM stiffness [11], proteolytic activity to melt down and rarefy the ECM in front of the cell [2, 5, 66], and nuclear damage in passages through small ECM gaps [67] are beyond the scope of the model. For example, in low-elasticity ECMs, cells migrate by developing nascent adhesions only, presumably applying very low traction forces in order not to deform the ECM too much [68]. What, if any, effect the cell nucleus has on migration in these cases is an open question.

The great modern surge in experimental research on 3D cell migration [4] has fittingly been accompanied by modelling studies that begin to address relevant mechanical questions [27, 62]. Very recently, modelling and experiments on 3D migration have merged [69–71]. Our study, hopefully, will contribute to the understanding of the general mechanical principles of 3D motility.

## Supporting information

**S1 Text. This file contains four sections.** Sections S1 and S2 describe the fluid dynamics and numerical methods used to solve the model equations. The remeshing algorithm used to simulate mechanism 2 is described in Section S3. Section S4 contains 16 figures showing the cell and ECM position at several time values for the parameter regimes listed in Table 1. An analysis of the sensitivity of the results to changes in nuclear size and bending rigidity of the cortex/nucleus is also described in Section S4.
(PDF)

**S1 Fig. Test on the remeshing algorithm with $N = 50$ points.** (a) Cell before remeshing. The back of the cell (red diamonds) is rigid with $k = 100$ pN/$\mu$m and $r = 0.5$, while the front of the cell (blue circles) is looser with $k = 1$ pN/$\mu$m and $r = 1$. (b) The arclength function $s(j)$ is sampled at equal distances to give the new node locations, which are interpolated from the old locations. (c) After remeshing, we note how the points are now evenly distributed. The point of interest in the error calculation is the first counter-clockwise red point in the second quadrant of the cell. (d) The new reference locations used to calculate the elastic force after remeshing.
(EPS)

**S2 Fig. Push-pull mechanism of motility through a sparse ECM.** The position of the cortex (green), nucleus (magenta), and ECM nodes (blue) at several time values during a simulation. Time values increase from left to right and top to bottom. Key parameter values for each simulation are indicated in the figure captions using bold text. See Table 1 for exact parameter values. The cell migrates easily through the sparse network (same as the network shown in Fig 1(a)).
(EPS)

**S3 Fig. Mechanism 1: $E > T$ and $N > T$.** The cell becomes stuck in the ECM during the simulation.
(EPS)

**S4 Fig. Mechanism 1: $T > E > N$.** The cell migrates by squeezing the nucleus, which is relatively more deformable than the cortex and ECM.
(EPS)

**S5 Fig. Mechanism 1: $T > N > E$.** The cell moves using a combination of nuclear and ECM deformation.
(EPS)

**S6 Fig. Mechanism 1: $N > T > E$.** The cell moves by a combination of nuclear and ECM deformation. However, in this parameter regime, the cell must deform the ECM to migrate because the nucleus is relatively stiffer (compare to the data in S7 Fig). Black $\times$'s are used to show the initial position of the key ECM nodes, which achieve their maximum displacement at $t = 4.53$.
(EPS)

**S7 Fig. Mechanism 1: $E > T > N$.** The cell moves by squeezing the nucleus because of relatively reduced tension in the cortex. Note the elongated cell shape for this parameter regime.
(EPS)

**S8 Fig. Mechanism 1: $N > T > E$ for a small nucleus.** We use the same parameters as in S6 Fig, but halve the diameter of the nucleus. The behavior is qualitatively more similar to the $T > N > E$ regime shown in S5 Fig. Thus halving the nuclear size simply decreases the characteristic force $N$ required to deform the nucleus.
(EPS)

**S9 Fig. Mechanism 1: Varying the bending energy in the parameter regime $E > T > N$.** (a) Simulation results with no bending energy (where the bending rigidity $K_b = 0$), and (b) results at comparable time values to (a) where the bending rigidity was increased to $K_b = 0.05$ throughout the entire nucleus, and in the cortex after binding to an ECM node. Distance traveled increases when bending is included for this parameter regime. The rounder nucleus in (b) results in a decrease in $N$ relative to $T$. The rounder cortex at the leading edge increases $T$, pulling the cell forward.
(EPS)

**S10 Fig. Mechanism 2: $E > T$ and $N > T$.** When the contraction of the cell cortex is too weak to deform either the nucleus or ECM, the cell is not able to squeeze through a gap in the ECM. (EPS)

**S11 Fig. Mechanism 2: $T > E > N$.** The cell can easily pass through the gap between the fibers because the nucleus is soft compared to the cortex and ECM. (EPS)

**S12 Fig. Mechanism 2: $T > N > E$.** The nucleus begins to buckle (bottom row) because of high cortical tension and increased cortical stiffness. After releasing the nodes, the nucleus partially regains its rounded shape (compare the position of the nucleus at $t = 5.18$ to $t = 4.14$). (EPS)

**S13 Fig. Mechanism 2: $N > T > E$.** In spite of the nucleus' relatively large stiffness, the cell is still able to migrate through the ECM. However, the cell is unable to entirely pass through the gap after one motility cycle. (EPS)

**S14 Fig. Mechanism 2: $E > T > N$.** Although the cell makes some progress, there is not enough force to drive it completely through the gap. After one motility cycle, the cell retracts slightly after releasing the nodes. (EPS)

**S15 Fig. Mechanism 2: $N > T > E$ for a small nucleus.** We use the same parameters as in S13 Fig, but halve the diameter of the nucleus. The behavior is qualitatively more similar to the $T > E > N$ regime shown in S11 Fig. (EPS)

**S16 Fig. Mechanism 2: Varying the bending energy in the parameter regime $E > T > N$.** (a) Simulation results with no bending energy (where the bending rigidity $K_b = 0$), and (b) results at comparable time values to (a) where the bending rigidity was increased to $K_b = 0.05$ in the stiff (rear) region of the cortex and throughout the entire nucleus. Increasing the strength of the force due to bending increases the nuclear force (represented by the parameter $N$) and prevents the cell from moving through the ECM. Results are similar to those from the regime $E > N > T$, where the cell was unable to migrate through the ECM. (EPS)

**S1 Table. Errors to show convergence of the remeshing algorithm.** Errors in the final position of the first red node in the second quadrant (in counterclockwise order). Test problem described in Section S3 of S1 Text. (XLSX)

**S2 Table. Mechanism 1: Distance traveled after the first cycle in S4–S8 Figs computed by tracking the nucleus' center of mass.** Penetration is calculated using the fraction of points on the nucleus and cortex that move past the line dividing the two ECM nodes that the cell moves past, approximately located at the points (0.5, ±0.5). Data indicated by * are simulated with a small nucleus ($r_{nuc} = 0.225$). (XLSX)

**S3 Table. Mechanism 1: Distance and penetration data with bending energy included on both the nucleus and cortex.** The bending rigidity $K_b = 0.05$, and the bending energy is computed using the preferred curvature of a circle. Bending energy on the cortex is only included after the cell binds to an ECM node, while it is always included on the nucleus. Percentages are

the percentage change from the data in S2 Table.
(XLSX)

**S4 Table. Mechanism 2: Distance traveled after the first cycle in S11–S15 Figs computed by tracking the nucleus' center of mass.** Penetration is calculated using the fraction of points on the nucleus and cortex that move past the line dividing the two ECM nodes approximately located at the points (0.5, ±0.5). Data indicated by * are simulated with a small nucleus ($r_{nuc} = 0.225$).
(XLSX)

**S5 Table. Mechanism 2: Distance and penetration data with bending energy included on both the nucleus and cortex.** The bending rigidity $K_b = 0.05$, and the bending energy is computed using the preferred curvature of a circle. Bending energy on the stiff part of cortex is only included after the cell binds to an ECM node, while it is always included on the nucleus. Percentages are the percentage change from the data in S4 Table.
(XLSX)

**S1 Video. Mechanism 1 through sparse ECM.** The cell migrates through a sparse ECM using mechanism 1 without deforming its nucleus.
(AVI)

**S2 Video. Failure for mechanism 1.** The cell becomes lodged in the ECM in simulations of the parameter regimes $E > N > T$ and $N > E > T$ using mechanism 1.
(AVI)

**S3 Video. Mechanism 2 through sparse ECM.** The cell migrates through a sparse ECM using mechanism 2 without deforming its nucleus.
(AVI)

**S4 Video. Failure for mechanism 2.** The cell becomes stuck in the ECM for parameter regimes $E > N > T$ and $N > E > T$ while migrating using mechanism 2.
(AVI)

**S5 Video. Nuclear buckling and relaxation.** A simulation of the parameter regime $T > N > E$ (using mechanism 2) shows the cell nucleus wrinkles, or buckles, under high tension in the rear. After the cell detaches from the ECM nodes, the nucleus relaxes, which induces a flow that inhibits the cell's forward progress through the ECM.
(AVI)

## Acknowledgments

We thank C. Copos for helpful discussions on the remeshing algorithm.

## Author Contributions

**Conceptualization:** Ondrej Maxian, Alex Mogilner, Wanda Strychalski.

**Data curation:** Ondrej Maxian, Wanda Strychalski.

**Formal analysis:** Ondrej Maxian, Alex Mogilner, Wanda Strychalski.

**Investigation:** Ondrej Maxian, Wanda Strychalski.

**Methodology:** Ondrej Maxian, Alex Mogilner, Wanda Strychalski.

**Project administration:** Alex Mogilner, Wanda Strychalski.

**Software:** Ondrej Maxian.

**Visualization:** Ondrej Maxian, Wanda Strychalski.

**Writing – original draft:** Ondrej Maxian, Alex Mogilner, Wanda Strychalski.

**Writing – review & editing:** Ondrej Maxian, Alex Mogilner, Wanda Strychalski.

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
