## [Decision Letter · Decision Letter 0]

5 Mar 2020

Dear %TITLE% Strychalski,

Thank you very much for submitting your manuscript "Computational estimates of mechanical constraints on cell migration through the extracellular matrix" for consideration at PLOS Computational Biology.

As with all papers reviewed by the journal, your manuscript was reviewed by members of the editorial board and by several independent reviewers. In light of the reviews (below this email), we would like to invite the resubmission of a significantly-revised version that takes into account the reviewers' comments.

We cannot make any decision about publication until we have seen the revised manuscript and your response to the reviewers' comments. Your revised manuscript is also likely to be sent to reviewers for further evaluation.

Sincerely,

Christopher V. Rao

Associate Editor

PLOS Computational Biology

Jason Haugh

Deputy Editor

PLOS Computational Biology

Reviewer's Responses to Questions

**Comments to the Authors:**

Reviewer #1: The article propose a computational model to describe cell migration in a 3D fibrous environment, which is different from the motion on a two-dimensional substratum, focusing in particular on the effect of several mechanical aspects, such as the contractile force acting on the cortex, the elasticity of the nucleus and the rigidity of the ECM. The main aim is to describe the difference between mesenchymal and ameboid modes of migration. This is a subject that is recently attracting a lot of attention.

In the abstract, the Authors' summary and the introduction it is stressed that the model is three-dimensional, so I was expecting a 3D cell. Actually, the computational environment is a slice of a three-dimensional set-up, as correctly stated at the beginning of the section on the qualitative model description. Justifying this simplification with axisymmetry arguments or indefinite extension in the third dimension is unrealistic. I think that the Authors should simply state that their computational domain is the mid-plane of the cell and, above all, anticipate the statement to the abstract and/or to the introduction.

Working in the framework of sub-cellular element models, the model consists in discretizing the cell and the nuclear membrane in a series of elements connected by spring with stiffness that can dynamically change according to the mode of migration. In this respect the work is similar to those by Bates' group (some cited) who however focus on bleb formation and don't devote much attention to the mechanical aspects of migration. On the other hand, the work by Scianna et al.

(Math. Biosci. Engng. 10, 235--261 (2013)) use a cellular Potts model focusing on the effect of the stiffness of the nucleus and of the extra-cellular matrix on cell migration mainly in a 3D environment with a regular and random distribution of ECM fibers. For this reason, I think that these results should be mentioned. I also find some similarities with the approach used by Schmeiser's group (see, for instance, J. Math. Biol. 74 (2017), pp. 169-193). So a comparison would be desirable.

Modelling

Generally speaking, I think that more details on the dynamics of the membrane should be given in the main text, so that the differences between the two migration modes can be more clearly related to the modelling terms and parameters.

ECM

The ECM is treated as a set of points connected by springs that in Eq.(1) seem to have vanishing rest length. An artificial addition of an extra-force (pinning-down force) is needed to ensure that the configuration is in equilibrium. Then, it is stated that this force is mimicking the effect of the resting length of the springs. Why not writing more correctly the mechanics of the springs with finite rest lengths given by the distance set between points? Then one could linearize the force to get Eq.(1). On the other hand, this would easily allow to introduce, if wanted, the highly nonlinear behaviour of ECM to strong deformations.

From the simulations it seems that the ECM actually reacts like a viscoelastic material (a sort of creep behaviour) rather than an elastic one. Where does this viscous behaviour comes from? From the cell fluid?

Mesenchymal migration

Regarding mesenchymal migration, some aspects of the dynamics should be described better, e.g., how is the protrusion generated? (randomly? Cells are not polarized and there is no chemotaxis, right? Or there is a predefined front and back?)

Is it possible to generate more protrusions as it actually occurs in cells? Why is this not done? In the movie successive spiking

Frankly speaking, the dynamics generated looks a bit strange because of the contemporary backward extension. Is this related to the fluid dynamics inside the cell?

Ameboid migration

Regarding ameboid migration, the fact that the two protrusions that are generated with an angle smaller than 45 degrees find anchoring points nearby depends on the location of ECM points. What happens if adhesion points are not found?

Is the stiffening homogeneous in the front and in the rear of the cell? Is it graded? Time-dependent? How is all this set?

How is the splitting between region identified?

What is distinguishing in the model the front pulling and the rear pushing? The fact that there is no f_0?

Minor points

Why the x-axis range between 0 and 4 and the y-axis range between -2 and 2.

Though an experimental article is cited, I'm really afraid that the volume of the cytoplasm is really small. There are pictures from Friedl's group related to 3D experiments where this volume is larger and in my opinion more realistic. Due to the compartimentalization between cytoplasm and nucleus, this might have strong impact on the fluid-dynamics inside the cytoplasm

Conclusion

I think that there are some points to be clarified and some more details on the modelling to be given, which requires major revisions before the article can be published.

Reviewer #2: The authors have developed a 2D computational framework to simulate cell migration through a fibrous environment by two different mechanisms: mesenchymal (push-pull) and amoeboid (rear-squeezing). The cell is divided into two compartment (nuclear and cytoplasmic), each is represented by a closed contour, and fluid flow is simulated in both of these compartments and in the extracellular environment using the boundary integral method (regularized Stokeslets in 2D, a standard approach for flow around microswimmers or obstacles). Because of the hydrodynamic equations, the contours cannot cross. Forces by nucleus (N), cell (T) and matrix (E) are represented by spring constants. The main results are that for successful migration, either T>N or T>E is required, but not both. More importantly, large N is a more severe limitation than large E. The authors also show that deformation energy stored in N or E as well as pressure in the fluid can aid migration.

The general conclusions drawn from this study are very interesting and certainly of interest to the large field working on cell migration. The paper is written and presented very well. The strong focus on nuclear mechanics of very timely. However, while the motivation and conclusions are clear and interesting, the actual model is not very transparent and has many issues. In the following, I list the main issues that I see.

Nucleus: nuclear size is fixed very close to cell size. The simulated cells do not look like real cells, the nucleus is very large and often appears to act more like a cortex. In fact it is modeled not as a bulk material, but as a hollow shell. There is also concern that the fluid flow between the two close-by contours is not represented well if they are so close, because then the cutoff function of the regularized Stokeslets might become relevant. In my view, the authors have to present also simulations with a smaller nucleus, because only then can the reader judge how well the method (in particular the hydrodynamics) works. They also should show some examples of fluid flow during deformations (not only the pressure field).

Volume conservation: it is not clear to me if and how volume (area in the 2D model) is kept constant.

Flow boundary conditions: it is not explained which boundary conditions result from the Stokeslet method (if any). Traditionally one would implement no-slip boundary conditions, at least in the normal direction. Can fluid mix between compartments ? Is pressure continuous across the boundaries as suggested by Fig. 5 ? What are the absolute values predicted by this theory and how does this compare to experimental results, e.g. in the context of blebbing ?

Viscosities: if we disregard the interaction of water with ECM-components, we expect an interstitial viscosity close to the one of water. Cytoplasmic viscosity on the scale of the cell is known to be many orders of magnitude higher than the one of water due to the presence of the cytoskeleton. Nuclear viscosity is expected to be even higher than the one of the cytoplasm due to high chromatin density. Thus I wonder why the three viscosities are all set to the same value here. Can the Stokeslet method be extended to different viscosity values and what would be the main consequences ?

Elasticity: cell and nuclear stiffness are represented by springs in the contour and there is neither a bulk elasticity nor a bending energy of the contour. This leads to many issues. The simulated shapes often have cusps that would not appear in real cells. I suggest to make also simulations with bending energy. Also there should be more discussion why bulk elasticity is not modeled and how realistic the conversion of bulk values to spring constants really is.

Matrix: to make the model more predictive, the authors should simulate different matrix densities and compare to literature results on how cells cope with intermediate versus high fiber densities (I agree that sparse is not interesting in this context). Maybe I have overlooked this, but it appears that density is not really varied, although this might be the most rewarding strategy to compare with experiments. Like in Figs. 3 and 4, it would be interesting to see cell speed as a function of density.

Comparison between the two migration modes: the authors find that the rear-squeezing mode is more efficient, but this might be mainly due to their specific rule that two rather than one protrusion is used here. What happened if the other mechanism is implemented with two protrusions ? How can the cell control protrusions at the front by a mechanism that acts at the back ? Obviously this mode requires more regulation than the push-pull mode. There is concern here that the rear-squeezing mode is better by construction.

Elastic forces: equation 3 is not clear to me. What is tau – the tangent vector ? The derivative of the tangent vector is the normal, so this force is always normal ?

There are a few word dublications in the text (“for a for a”, “the the”).

Reviewer #3: The review is uploaded as an attachment

**Have all data underlying the figures and results presented in the manuscript been provided?**

Reviewer #1: No: Data to get the simulations are incomplete

Reviewer #2: Yes

Reviewer #3: Yes

PLOS authors have the option to publish the peer review history of their article (what does this mean?). If published, this will include your full peer review and any attached files.

Reviewer #1: No

Reviewer #2: No

Reviewer #3: No
---

## [Decision Letter · Decision Letter 1]

26 Jun 2020

Dear %TITLE% Strychalski,

Thank you very much for submitting your manuscript "Computational estimates of mechanical constraints on cell migration through the extracellular matrix" for consideration at PLOS Computational Biology. As with all papers reviewed by the journal, your manuscript was reviewed by members of the editorial board and by several independent reviewers. The reviewers appreciated the attention to an important topic. Based on the reviews, we are likely to accept this manuscript for publication, providing that you modify the manuscript according to the review recommendations. In particular, Reviewer 1 was satisfied with the revisions while reviewer 2 still questions some assumptions.

Sincerely,

Christopher V. Rao

Associate Editor

PLOS Computational Biology

Jason Haugh

Deputy Editor

PLOS Computational Biology

[LINK]

Reviewer's Responses to Questions

**Comments to the Authors:**

Reviewer #1: nothing

Reviewer #2: The authors have performed major revisions on their manuscript. In my view, they have responded very well to most technical questions and the revised work is in very good shape regarding the continuum mechanics part. From the biological point of view, I appreciate the demonstration that relative stiffness of nucleus and matrix is very important for cell migration, and that hydrodynamic pressure might play a large role, favoring one mode of cell migration over another. However, I also think that the response is not sufficient yet in regard to the biological system of interest. In particular, I think that there are still two major issues that have to be addressed.

First they now write in the introduction “One represents the nucleus which is sometimes almost as great in size as the whole cell [31]” and later “The cell diameter is of the order 10 um, while the nuclear diameter is slightly smaller [31].” I checked Ref. 31, the PNAS paper 2009 from the Paluch group, they use fibroblasts and the nucleus here is expected to be much smaller than the cell. Actually there seem to be no data on this in that paper, but compare the cartoon in the supplement and the general literature on fibroblasts. I think that the authors should go back to the literature and give evidence for such large nuclear sizes (in my understanding, this is mainly true for stem cells, but it would also be interesting to check for white blood cells). Then they should clearly state that most model cell lines have larger cytoplasms and include new Figs. S8 and S15 (which look much more realistic than the current main figures with the large nuclei) into the main text with discussion, because this important subject should not be relegated to the supplement.

Second the authors now write “The nucleus and cortex have an elastic energy, which represents a bulk modulus in three-dimensions [33].” I am not sure if I agree on this. Their elastic energy seems to be a stretching energy or cortical tension (a spring constant or line tension in 2D, surface stretching energy or surface tension in 3D), but not a real bulk energy like in a fully elastic solid. That is fine with me as a modelling choice, but should be clarified. Of course a stretching energy can be related to a 3D modulus in a thin film approximation, but this is not the point. I think the cortical buckling also comes from the fact that the authors effectively use thin shell elasticity and hydrodynamics (if the 2D model was transferred to 3D), but not bulk elasticity. It is a clear modelling choice to represent the cell by two elastic contours. Maybe I misunderstand something here, but the text is not clear yet on these important issues.

**Have all data underlying the figures and results presented in the manuscript been provided?**

Reviewer #1: None

Reviewer #2: None

PLOS authors have the option to publish the peer review history of their article (what does this mean?). If published, this will include your full peer review and any attached files.

Reviewer #1: No

Reviewer #2: No
---

## [Editor Report · Decision Letter 2]

17 Jul 2020

Dear %TITLE% Strychalski,

We are pleased to inform you that your manuscript 'Computational estimates of mechanical constraints on cell migration through the extracellular matrix' has been provisionally accepted for publication in PLOS Computational Biology.

Best regards,

Christopher V. Rao

Associate Editor

PLOS Computational Biology

Jason Haugh

Deputy Editor

PLOS Computational Biology
